# Infralimbic medial prefrontal cortex signalling to calbindin 1 positive neurons in posterior basolateral amygdala suppresses anxiety- and depression-like behaviours

Huiling Yu [1], Liping Chen[2], Huiyang Lei [1], Guilin Pi [1], Rui Xiong [1], Tao Jiang [1], Dongqin Wu [1], Fei Sun[1], Yang Gao [1], Yuanhao Li[1], Wenju Peng[1], Bingyu Huang[1], Guoda Song[1], Xin Wang[1], Jingru Lv[1], Zetao Jin[1], Dan Ke[1], Ying Yang [1] ✉ & Jian-Zhi Wang [1,3] ✉

Generalization is a fundamental cognitive ability of organisms to deal with the uncertainty in real-world situations. Excessive fear generalization and impaired reward generalization are closely related to many psychiatric disorders. However, the neural circuit mechanism for reward generalization and its role in anxiety-like behaviours remain elusive. Here, we found a robust activation of calbindin 1-neurons (Calb 1) in the posterior basolateral amygdala (pBLA), simultaneous with reward generalization to an ambiguous cue after reward conditioning in mice. We identify the infralimbic medial prefrontal cortex (IL) to the pBLA$^{Calb1}$ (Calb 1 neurons in the pBLA) pathway as being involved in reward generalization for the ambiguity. Activating IL−pBLA inputs strengthens reward generalization and reduces chronic unpredictable mild stress-induced anxiety- and depression-like behaviours in a manner dependent on pBLA$^{Calb1}$ neuron activation. These findings suggest that the IL−pBLA$^{Calb1}$ circuit could be a target to promote stress resilience via reward generalization and consequently ameliorate anxiety- and depression-like behaviours.

Different emotional preferences, such as optimistic and pessimistic preferences, often arise in our daily life, especially under uncertainty. When facing ambiguous situations, the optimistic individuals often show full of enthusiasm and keep trying with good expectations. However, the pessimistic individuals always negatively distort the interpretation of the ambiguous information and show extensive avoidance[1–3], which is particularly evident in anxiety and depression[1–3]. In addition to heritability, these distinct emotion biases to the ambiguity may be shaped by the generalization from experienced events. Therefore, correcting excessive aversion generalization and/or

improving safe/reward generalization may be efficient strategy to ameliorate anxiety- and depression-like behaviours.

The basolateral amygdala (BLA), including the lateral (LA) and basal (BA) nuclei of the amygdala[4], receives sensory information from multiple modalities[5–10]. It not only translates certain cues into motivated behaviours[10,11], but also participates in the generalization of emotional memory[12]. In fear conditioning, a neutral stimulus (a conditional stimulus, CS) is definitely paired with an aversive outcome such as a mild foot shock. Upon this cue encoding, BLA neurons activate and trigger fear-associated freezing behaviour[7,13,14]. Intriguingly,

[1]Department of Pathophysiology, School of Basic Medicine, Key Laboratory of Education Ministry of China/Hubei Province for Neurological Disorders, Tongji Medical College, Huazhong University of Science and Technology, Wuhan 430030, China. [2]Department of Gastroenterology, Tongji Hospital, Tongji Medical College, Huazhong University of Science and Technology, Wuhan, China. [3]Co-innovation Center of Neuroregeneration, Nantong University, Nantong 226000, China. ✉e-mail: yingyang@hust.edu.cn; wangjz@mail.hust.edu.cn

when the intensity of unconditioned stimulus increases, some LA neurons that respond to CS begin to lose their cue specificity and activate simultaneously with generalization of conditioned fear[15]. After increasing neuronal excitability of LA neurons, generalized fear is remarkably produced, indicating the role of LA in the formation of fear generalization[15]. In addition to fear and generalized fear, BLA has also been implicated in safety-[16–18] and reward-[7,19–21] related behaviours. The presence of a compound light-tone cue that accurately predicts reward can robustly activate BLA neurons[7]. Targeting these BLA neurons remarkably supports positive reinforcement[7]. Given the impairment of reward generalization on some individuals with pathological anxiety[22] and the anxiolytic effect of the posterior part of BLA (pBLA)[23], it will be valuable to explore whether and how the distinct pBLA neurons with their associated circuits necessarily lead to reward generalization and promote stress resilience to ameliorate anxiety- and depression-like behaviours.

The infralimbic (IL) region of the medial prefrontal cortex (mPFC) is one of the upstream regions of the BLA[24], and is involved in both negative[25] and positive[26,27] emotional behaviours. Electrostimulation of IL dramatically suppresses conditioned fear responses[28] and facilitates fear extinction[29], while inactivation of IL impairs consolidation and the retrieval fear extinction[30]. Cue-induced drugs and natural rewards could also activate IL[26]. Nonselective disruption of IL prevents reward seeking-associated promotion[31,32], indicating the role of IL as an integration hub for reward seeking behaviours. For generalization, inhibition of prefrontal inputs into the N. reuniens is sufficient to produce overgeneralization of fear memories[12]. When using a partially reinforced fear paradigm, enhanced fear generalization is observed after inhibition of mPFC→BNST pathway[33]. However, whether IL communicates with pBLA and how they orchestrate to generalize reward and modulate anxious states have not been reported.

A go/no-go task was previously introduced in rodents to investigate behaviour responses when facing an ambiguous cue[34]. In this paradigm, rats were trained to press a lever in the presence of a tone associated with a positive event (delivery of a food pellet) and to refrain from pressing the lever as a way to avoid a negative event (white noise) associated with another tone. By using ambiguous tone to probe animals' relative anticipation of positive or negative events, few and slower lever presses were found on rats that experienced unpredictable stress, indicating reduced anticipation of a positive event. It is worth noting that this paradigm cannot distinguish whether a response bias originates from a reduced positive and/or an increased negative response. Moreover, a 'no-go' as a response indicator cannot be distinguished from a response omission. Inspired by platform-mediated avoidance[35–37] and discriminative conditioning[16,38], we developed a new go/go task in the present study. We first trained the mice to poke for sucrose reward associated with one tone and to step on an insulated platform in response to a different tone to avoid foot-shock punishment. Then, the mice were tested for their responses to an ambiguous probe tone of intermediate frequency. Finally, the expectations of a positive or negative event signalled by ambiguous tone were inferred from their active poking or step-on responses.

By combining our novel go/go-task behavioural tests with neural circuit-specific manipulations, we found reward generalization in the presence of ambiguous cues after reward conditioning (RC). Interestingly, -50% of mice showed more reward-approaching behaviours than aversion-avoidance behaviours in response to ambiguous probe tones after experiencing different tone-paired reward and aversive training. The pBLA[Calb1] neurons and the IL to pBLA inputs were robustly activated in mice with high reward generalization (HRG). Inhibiting pBLA[Calb1] neurons or IL−pBLA inputs significantly attenuated reward generalization for ambiguous cues, while photoactivation produced opposite effects. Importantly, stimulating IL−pBLA inputs dramatically ameliorated anxious and depressive phenotypes of chronic unpredictable mild stress (CUMS) mice in dependence of pBLA[Calb1] neuron

activation. To the best of our knowledge, this is the first identification of IL−pBLA[Calb1] circuit in reward generalization, which could be a potential target for ameliorating anxiety- and depression-like behaviours.

## Results

### High reward generalization for ambiguous cues is associated with robust activation of pBLA[Cab1] neurons

To investigate reward generalization in the presence of an ambiguous information, we first established a reward conditioning (RC) mouse model by training the mice to associate a 2000 Hz tone (conditioning stimulation, CS) with sucrose. During 16 days of training, an association between accurate cues (2000 Hz tone) and sucrose reward was gradually established (Fig. 1a, b). At Day 17, the success of the RC model was validated by the significantly increased sucrose poking time in response to CS tone, compared with the Pre-CS (Fig. 1c). Furthermore, the RC was also shown regardless of whether the same cohort of mice were in their training box or in a novel box (Fig. 1c).

Then, we delivered an ambiguous tone (5000 Hz) to these CS-trained mice to test their reward generalization for ambiguous cues (Fig. 1d). The mice began to poke when the ambiguous tones were delivered, indicating the generalization of reward memory; but with fewer poking times and longer first poking latency to the ambiguous tone than the CS tone (Fig. 1e, f), indicating the ability to distinguish the ambiguous cue from the CS tone. Further analysis based on their poking times in response to the ambiguous tone revealed that the mice could be divided into two groups, i.e., high reward generalization (HRG) and low reward generalization (LRG) groups (Fig. 1g). The HRG mice had more while the LRG mice had fewer poking times in response to the ambiguous tone, and both groups had similar poke times in response to the CS tone (Fig. 1g). By c-Fos staining, we observed that activation of emotion- and decision-associated brain regions, i.e., pBLA, the posterior part of the basomedial amygdala (BMP) and IL, was greater in HRG mice than in LRG ones in response to the ambiguous tone (Fig. 1h, i); furthermore, the c-Fos signals in the pBLA were largely colocalized with calbindin1−positive neurons (pBLA[Calb1]) in the HRG group (Fig. 1j–l). Together, these data indicate that activation of pBLA[Calb1] neurons is involved in the reward generalization for ambiguity.

By developing a go/go-task paradigm, we further investigated the association of pBLA[Calb1] neurons and reward generalization for the ambiguity in mice that experienced both reward and aversive conditioning (Fig. 2a). In this paradigm, the mice were first trained to associate the sucrose reward with 2000 Hz tones (Fig. 2b) and foot-shock punishment with 9000 Hz tones (Fig. 2c and Supplementary Fig. 1). These positive and negative conditioning tones were counterbalanced across animals. After RC and Fear conditioning training (FC), the CS-triggered step-on response in the FC test and CS-induced poking response in the RC test could be well established in mice (Fig. 2d, e). When an intermediate tone (5000 Hz) was first delivered as an ambiguous cue, 50% of mice preferred poking (reward preference, termed as RP), while 45% of mice chose to step on the platform (aversion preference, termed as AP) (Fig. 2f). This first choice was positively correlated with the next 9 choices during the ambiguous tone probing (Supplementary Fig. 2). Again, the activation of pBLA[Calb1] neurons was more significant in the RP group than in the AP group (Fig. 2g, h–l) with no change in pBLA[Calb1] neuron numbers (Fig. 2j). To further investigate the valence-associated activation of pBLA[Calb1] neurons, we designed a social reward to replace the sucrose reward in the go/go paradigm (Supplementary Fig. 3a). The mice were trained to poke to gain access to and interact with a target animal when the 2000 Hz tone was on (positive CS tone, Supplementary Fig. 3b). Then, the same cohort of mice was subjected to FC training (negative CS tone, 9000 Hz, Supplementary Fig. 3c, d). Tests on specific tones were conducted and confirmed the success of these

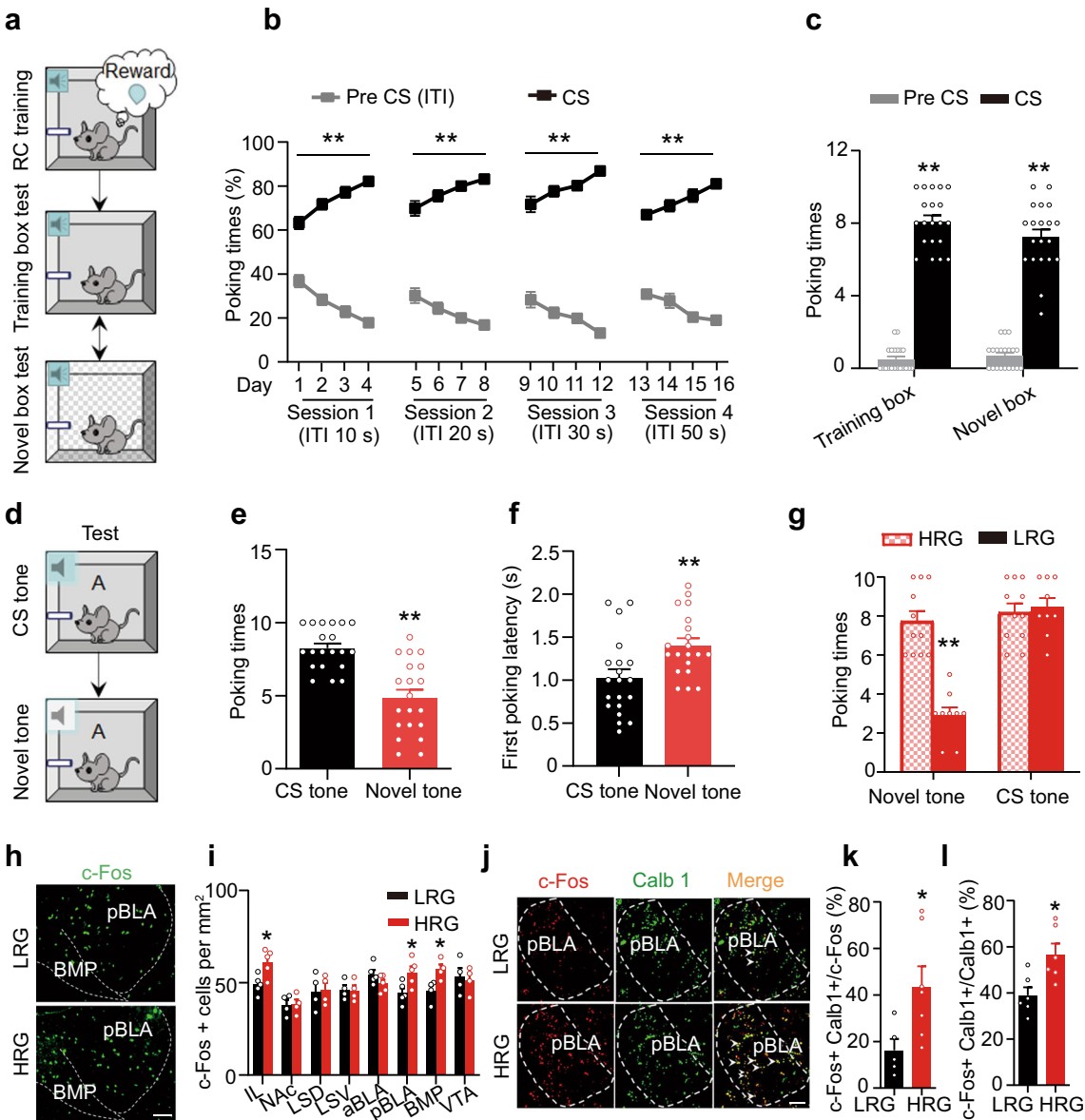

**Fig. 1 | High reward generalization for ambiguous cues is associated with robust activation of pBLA$^{Calb1}$ neurons. a** Schematic of reward conditioning (RC) training and test. **b** RC training improved association of CS tone with predicted sucrose. ITIs: inter-trial intervals. Repeated measures two-way ANOVA, [Session 1]: F(3, 57) = 56.40, $P < 0.0001$; [Session 2]: F(3, 57) = 31.82, $P < 0.0001$; [Session 3]: F(3, 57) = 30.49, $P < 0.0001$; [Session 4]: F(3, 57) = 33.63, $P < 0.0001$,Tukey's multiple comparisons test, **$P < 0.01$ vs Pre CS (ITI). **c** CS tones induced an increased poking time in both training and novel box. $n = 20$ mice per group. [Training box]: $t = 18.7$, $df = 19$, $P < 0.0001$; [Novel box]: $t = 15.38$, $df = 19$, $P < 0.0001$. **$P < 0.01$ vs Pre CS. **d** Schematic for ambiguous cue test. **e**, **f** CS-trained mice displayed generalization to ambiguous cues. $n = 20$ mice per group. [Poking times]: $t = 4.876$, $df = 19$, $P < 0.001$; [First poking latency]: $t = 3.370$, $df = 19$, $P = 0.0032$. **$P < 0.01$ vs CS tone.

**g** CS-trained mice were divided into high (HRG) and low (LRG) reward generalization groups. Cutoff: 5 poking times to ambiguous tones (left). HRG and LRG groups had same poking times to CS tones (right). $n = 11$ (HRG) or 9 (LRG) mice. [Novel tones]:$t = 6.959$, $df = 18$, $P < 0.0001$; [CS tones]:$t = 0.3927$, $df = 18$, $P = 0.6996$. **$P < 0.01$ vs HRG. **h**, **i** c-Fos staining and quantitative analysis. Scale bar, 100 μm. $n = 4$ or 5 mice per group. [IL]:$t = 3.056$, $df = 8$, $P = 0.0157$; [pBLA]: $t = 2.683$, $df = 8$, $P = 0.0278$; [BMP]: $t = 2.672$ $df = 6$, $P = 0.0369$. *$P < 0.05$ vs LRG. **j**–**l** Activation of pBLA$^{Calb1+}$ neurons was more in HRG than LRG group. Scale bar, 100 μm. LRG: $n = 5$ (**k**) or 6 (**l**) mice, HRG: $n = 7$ (**k**) or 6 (**l**) mice. $t = 2.440$, $df = 10$, $P = 0.0348$ (k); $t = 2.999$, $df = 10$, $P = 0.0134$ (l). *$P < 0.05$ vs LRG. Two-sided paired $t$ test was employed in **c**, **e**–**g**, **i**, **k**, **l**. Data were presented as mean ± SEM. Source data are provided as a Source Data file.

RC and FC training (Supplementary Fig. 3e, f). When an intermediate tone (5000 Hz) was delivered, 25% of mice preferred poking for social interaction (reward preference, RP(S)), while 70% of mice chose to step on the platform (aversion preference, AP(S)) (Supplementary Fig. 3g). Again, the pBLA$^{Calb1}$ neurons in the RP (S) group were more significantly activated than those in the AP (S) group (Supplementary Fig. 3h–j) without changes in pBLA$^{Calb1}$ neuron numbers (Supplementary Fig. 3k). These data provide further evidence supporting the involvement of pBLA$^{Calb1}$ neurons in reward generalization for ambiguity.

## pBLA$^{Calb1}$ neurons govern reward generalization for ambiguous cues

To identify the causal role of pBLA neurons, especially its Calb1+ neurons, in the reward generalization, we first employed chemogenetic manipulation on pBLA (Supplementary Fig. 4) and pBLA$^{Calb1}$ (Supplementary Fig. 5) neurons in HRG and LRG mice, respectively. Excitatory (AAV8-CaMKIIα-hM3Dq-eYFP or AAV8-DIO-hM3Dq-eYFP; hM3Dq) and inhibitory (AAV8-CaMKIIα-hM4Di-eYFP or AAV8-DIO-hM4Di-eYFP; hM4Di) chemoreceptors were bilaterally injected into the pBLA of C57BL/6 mice (Supplementary Fig. 4a–c, f–h) or Calb1-

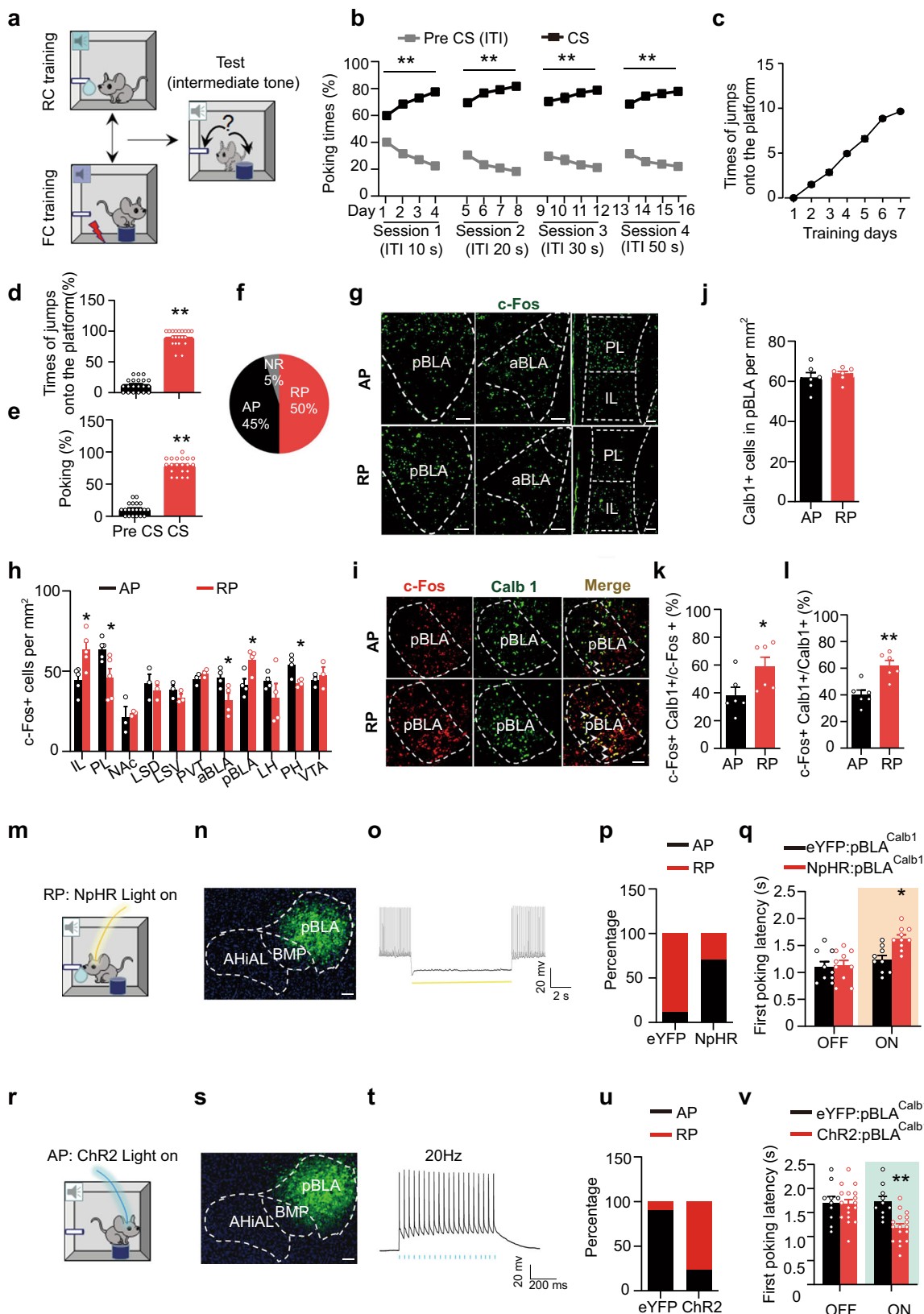

IRES-Cre mice (Supplementary Fig. 5a–c, f–h). In hM4Di-HRG group, administration of clozapine-N-oxide (CNO, 10 mg kg⁻¹ in saline) significantly decreased the poking rate (Supplementary Figs. 4d, 5d) and increased the first poking latency (Supplementary Figs. 4e, 5e) in response to the intermediate tone. While, hM3Dq-LRG mice showed a higher poking rate (Supplementary Figs. 4i and 5i) and shorter first

poking latency (Supplementary Figs. 4j and 5j) after CNO injection when the intermediate tone was delivered. These data indicate a causal role of pBLA neurons, especially its Calb1+ neurons, in reward generalization for the novel ambiguous cue.

Then, we applied an optogenetic method by expressing ChR2 and NpHR selectively in pBLA^Calb1 neurons, and then modulated their

**Fig. 2 | pBLA^Calb1 neurons govern reward generalization for ambiguous cue.**
**a** Schematic of a go/go paradigm. **b** Mice learnt to associate the positive CS tones with sucrose during RC training. $n = 20$ mice per group. Repeated measures two-way ANOVA, [Session 1]: $F(3, 57) = 88.46$, $P < 0.0001$; [Session 2]: $F(3, 57) = 33.64$, $P < 0.0001$; [Session 3]: $F(3, 57) = 7.35$, $P < 0.0001$; [Session 4]: $F(3, 57) = 14.72$, $P = 0.0005$, Tukey's multiple comparisons test, **$P < 0.01$ vs Pre CS (ITI). **c** Mice learnt to associate the negative CS tones with foot shocks during FC training. $n = 20$ mice per group. **d, e** Mice showed correct discrimination behaviours in FC (**d**) and RC test (**e**). $n = 20$ mice per group. [Times of jumps onto the platform%]: $t = 19.09$, $df = 19$, $P < 0.0001$; [Poking%]: $t = 16.49$, $df = 19$, $P < 0.0001$. **$P < 0.01$ vs Pre CS. **f** Proportion of reward preference (RP), aversion preference (AP) and no response (NR) mice. **g, h** c-Fos staining and quantitative analysis. Scale bar, 100 μm. $n = 3$–5 mice per group. [IL]: $t = 3.153$, $df = 8$, $P = 0.0135$; [PL]: $t = 2.773$, $df = 8$, $P = 0.0242$; [aBLA]: $t = 2.636$, $df = 6$, $P = 0.0388$; [pBLA]: $t = 2.916$, $df = 6$, $P = 0.0268$; [PH]:

$t = 3.285$, $df = 6$, $P = 0.0167$. *$P < 0.05$ vs AP. **i–l** Co-staining and quantitative analysis. Scale bar, 100 μm. $n = 6$ mice per group. $t = 0.6014$, $df = 10$, $P = 0.5610$ (**j**); $t = 2.294$, $df = 10$, $P = 0.0447$ (**k**); $t = 4.220$, $df = 10$, $P = 0.0018$ (**l**). *$P < 0.05$ vs AP.
**m, r** Schematic of optogenetic stimulations. **n, s** Representative images of injection site. Scale bar, 100 μm. **o, t** Illumination inhibited NpHR cells (**o**), while activated ChR2 cells in the pBLA (**t**). **p, q, u, v** Effects of pBLA^Calb1 manipulation on the RP proportion and the first poking latency. Two-way ANOVA, [**p, q**] $n = 9$ (eYFP:p-BLA^Calb1) or 10 (NpHR:pBLA^Calb1) mice, $F(1, 34) = 4.744$, $P = 0.0107$, Tukey's multiple comparisons test, *$P < 0.05$ vs pBLA: eYFP light on; [**u, v**] $n = 10$ (eYFP:pBLA^Calb1) or 17 (ChR2:pBLA^Calb1) mice, $F(1, 50) = 6.897$, $P = 0.0017$, Tukey's multiple comparisons test, **$P < 0.01$ vs pBLA: eYFP light on. Two-sided paired $t$ test was employed in **d, e, h, j–l**. Data were presented as mean ± SEM. Source data are provided as a Source Data file.

activity simultaneously with the intermediate tone in the go/go paradigm (Supplementary Fig. 6 and Fig. 2m–o, r–t). In the RP group, stimulation of NpHR-expressing pBLA^Calb1 neurons significantly prolonged the first poking latency, indicating the necessity of pBLA^Calb1 neurons in reward generalization for ambiguous cues (Fig. 2p, q). Conversely, activation of ChR2-expressing pBLA^Calb1 neurons shortened the latency for the first poking in AP group (Fig. 2u, v). These data demonstrate that the pBLA^Calb1 neurons are required and sufficient to govern reward generalization for the ambiguity.

## IL−pBLA inputs trigger reward generalization for the ambiguity

To further explore the neural circuit related to the reward generalization, we stereotaxically injected a retrograde tracer (CTB, Cholera toxin subunit B) into the pBLA (Fig. 3a). Robust signals were clearly shown in the IL (Fig. 3b). This finding was further supported by HSV anterograde tracing (Fig. 3c, d). These data indicate a physical connection between the IL and pBLA. To further verify whether there is a direct monosynaptic functional connection between IL and pBLA^Calb1 neurons, we used ex vivo brain slice recording. Upon stimulating IL−pBLA inputs in the pBLA, robust responses were recorded from Calb1+ neurons in the pBLA (Fig. 3e, f). Then, Tetrodotoxin (TTX, 1 μM) and 4-amynopyridine (4AP, 100 μM) were employed in the bath to remove any network activity. In IL−pBLA−ChR2 mice, the light-induced EPSC amplitudes in pBLA^Calb1 neurons persisted after TTX + 4AP perfusion (Fig. 3f, g). These data confirmed a direct and monosynaptic excitatory inputs from the IL to pBLA Calb1+ neurons.

Then we analyzed the firing activity of the IL−pBLA circuit during decision-making in response to ambiguous cues in freely moving mice. GCaMP6f was precisely expressed in IL neurons, which were shown to innervate pBLA neurons by injecting rAAV-hSyn-Cre into the pBLA and AAV-EF1a-DIO-GCaMP6f into the IL (Fig. 3h, i). When the intermediate tone was on, an increased calcium activity accompanied by poking was detected in the IL−pBLA circuit in both RP (Fig. 3j, l) and AP (Fig. 3k, m) mice, and importantly, the circuit activation appeared before poking behaviour (Fig. 3j–n). The average peak calcium activity and the slope for calcium transit before poke increased more significantly in the RP group than in the AP group (Fig. 3n, o). No significant fluctuation in calcium activity was detected in IL−pBLA-GFP controls (Fig. 3n and Supplementary Fig. 7). These data suggest that the activation of the IL−pBLA circuit controls reward generalization for the ambiguity, and that the faster and stronger the activation of the IL−pBLA circuit is, the higher reward generalization for ambiguity.

To further verify whether optogenetic manipulation of the IL−pBLA pathway controls reward generalization, we infused AAV8-CaMKIIa-eNpHR3.0-eYFP or AAV8-CaMKIIa-hChR2(H134R)-eYFP into the IL and implanted bilateral optical fibres in the pBLA to target IL−pBLA terminals by light stimulation (Fig. 3p, q, t and Supplementary Fig. 8). During the light-on epoch, the reward preference mice in the IL−pBLA-NpHR group exhibited fewer poking responses to the first intermediate tone (Fig. 3r) and longer first poking latencies compared

with IL−pBLA-eYFP control mice (Fig. 3s), indicating an inhibited reward preference by blocking IL−pBLA inputs. On the other hand, light stimulation of aversion-preferred mice in the IL−pBLA-ChR2 group dramatically increased the poking response to the first intermediate tone (Fig. 3u) and decreased the poking latency (Fig. 3v), suggesting an increased reward preference in response to ambiguous cues by stimulating IL−pBLA inputs. No significant changes in aversive preference were observed in IL−pBLA-NpHR-AP group (Supplementary Fig. 9). Given the anatomical connection between PL and pBLA in physical condition (Fig. 3b), we infused AAV8-CaMKIIa-hChR2(H134R)-eYFP into the PL of AP group (Supplementary Fig. 10) and manipulated PL−pBLA inputs to investigate its role in reward generalization. For PL−pBLA-ChR2-AP mice, the first latency to poke was not significantly changed upon photostimulation (Supplementary Fig. 11a–c). Also, no significant change of first poking latency was observed in PL−pBLA-NpHR-RP group as compared with controls (Supplementary Fig. 11d–f). Therefore, these data demonstrated that the IL−pBLA circuit, not PL-pBLA circuit, controls reward generalization for the ambiguity and triggers the reward-related approaching behaviours.

## Activation of the IL−pBLA^Calb1 circuit ameliorates anxiety- and depression-like behaviours in CUMS mice

Excessive negative generalization for ambiguity is one of the characteristics of anxiety- and depression-like behaviours[39–41]. Interestingly, we observed that the anxiety- and depression-like behaviours were negatively correlated with reward generalization for an ambiguous cue in the go/go paradigm (Supplementary Fig. 12a, b). Furthermore, compared with LRG mice, HRG mice displayed an increased resistance to CUMS (produced by 5-week treatment), evidenced by relatively more centre zone exploration in the open field test (OFT, Supplementary Fig. 13a–c), more open-arm exploration in the elevated plus maze (EPM, Supplementary Fig. 13d–f), and greater sucrose consumption in the sucrose preference test (SPT, Supplementary Fig. 13g, h) compared with LRG mice.

To explore whether and how the reward generalization-associated IL−pBLA^Calb1 circuit is involved in anxiety- and depression-like behaviours, we modified the CUMS paradigm and established a mouse model with anxiety/depression-phenotype (Fig. 4a). During the OFT, EPM, and SPT, the mice that experienced CUMS showed fewer centre entries and shorter time spent in the centre in OFT (Fig. 4b–d), fewer open-arm entries and less time spent in the open arm in EPM (Fig. 4e–g), and lower sucrose preference (Fig. 4h, i) in SPT. By CTB retrograde tracing and c-Fos co-staining, we found that the CUMS mice had fewer c-Fos+ and CTB+ neurons in the IL than the controls (Fig. 4j, k). No significant Calb1+ neuron loss was found in the CUMS group (Supplementary Fig. 14). These data indicate an inhibition of the IL−pBLA circuit in CUMS mice. During the illumination epoch, the IL−pBLA-ChR2-CUMS mice exhibited more open-arm and centre exploration, with increased approaches during the EPM and OPT, respectively (Fig. 5a–g). They also showed higher sucrose preference in

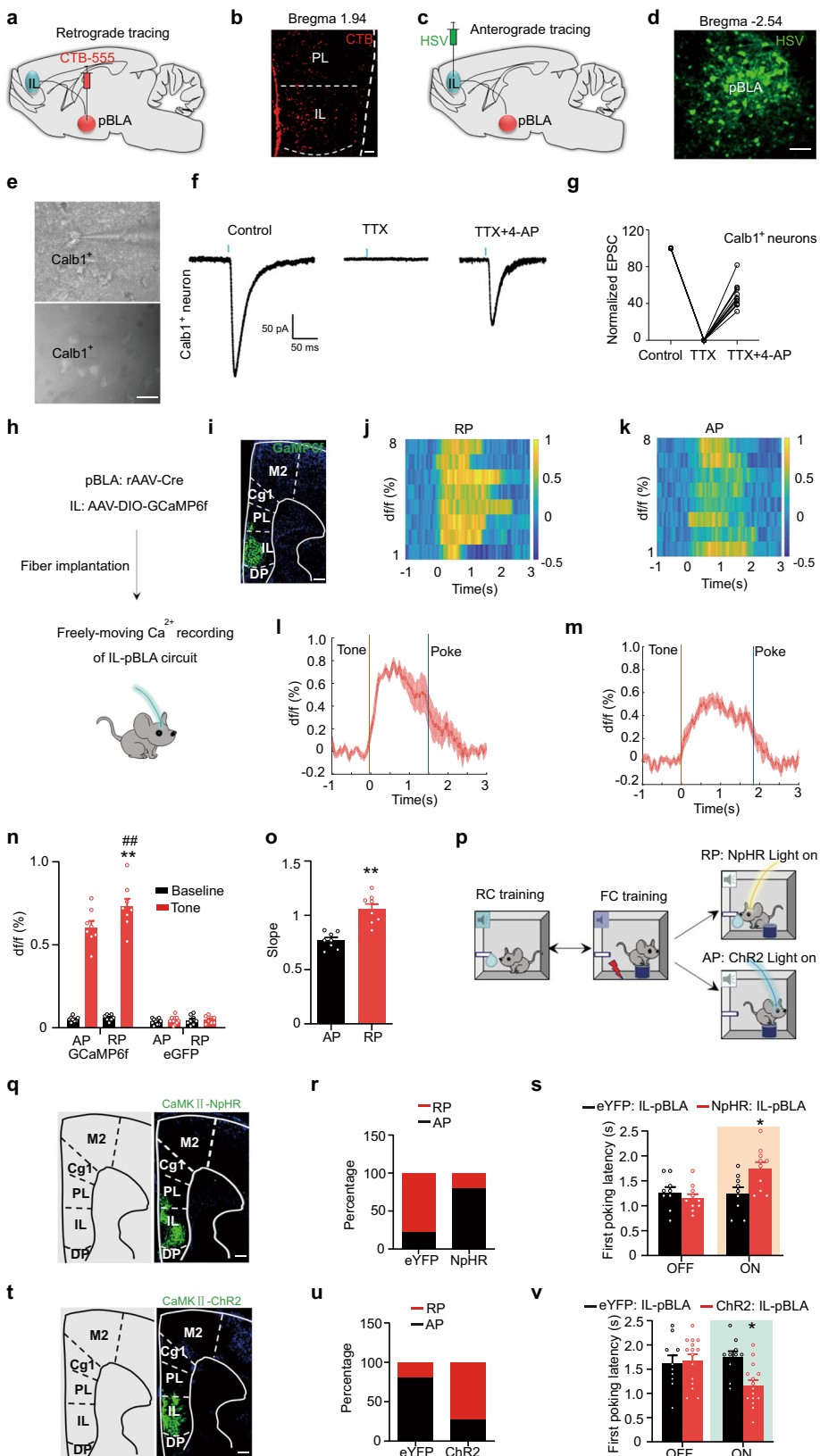

SPT (Fig. 5a, h). Together, these data demonstrate that the activating IL−pBLA circuit is sufficient to promote stress resilience and consequently ameliorate anxiety- and depression-like behaviours.

Given that pBLA$^{Calb1}$ neurons govern a reward generalization for the ambiguity (Supplementary Fig. 5 and Fig. 2m−v), we then asked whether pBLA$^{Calb1}$ neurons may mediate the suppression of anxiety-

and depression-like behaviours by the IL−pBLA pathway. To inactivate pBLA$^{Calb1}$ neurons and simultaneously activate IL−pBLA inputs, we injected AAV8-CaMKIIa-hChR2(H134R)-eYFP and AAV-DIO-hM4Di into the IL and pBLA, respectively, in Calb1-Cre-CUMS mice (Fig. 5a). Optical fibres were bilaterally implanted in the pBLA to target IL−pBLA terminals by delivering light (Fig. 5a). When CNO (10 mg/kg) was

**Fig. 3 | IL–pBLA inputs trigger reward generalization for the ambiguity.**
**a–d** Schematic and representative image of CTB retrograde tracing (**a**, **b**) and HSV anterograde tracing (**c**, **d**). scale bar = 100 μm (**b**) or 50 μm (**d**). **e** Representative images of patch pipette tips on pBLA$^{Calb1+}$ neurons in brain slices (scale bar = 30 μm). **f**, **g** Representative traces (**f**) and amplitude (**g**) of EPSCs in the IL–pBLA$^{Calb1+}$ pathway recorded under different experimental conditions. n = 13 pBLA$^{Calb1+}$ neurons from 6 IL–pBLA–ChR2 mice. **h** Schematic of Ca$^{2+}$ recording of IL–pBLA circuit. **i** Representative image of GCaMP6 expression in IL neurons projecting to the pBLA (IL–pBLA neurons). Scale bar, 200 μm. **j**, **k** Heat maps of normalized Ca$^{2+}$ activity in IL–pBLA neurons when RP (**j**) and AP (**k**) mice conducted poking behaviour in response to intermediate tones. **l**, **m** Average Ca$^{2+}$ transients from IL–pBLA neurons in response to intermediate tones (n = 8 mice per group). **n** Average df/f for baseline and peak df/f in response to intermediate tones. Two-way ANOVA, n = 8 (GCaMP6-AP, GCaMP6-RP, eGFP-RP), or 10 (eGFP-AP) mice. F(3,60) = 138.1,

$P < 0.0001$, Bonferroni post hoc analysis, \*\*$P < 0.01$ vs Baseline, ## $P < 0.01$ vs AP. **o** Rise slopes for calcium transits. Two-sided unpaired $t$ test, $t = 5.347$, $df = 14$, $P = 0.0001$, n = 8 mice per group. \*\*$P < 0.01$ vs AP. **p** Schematic of optogenetic stimulations in go/go paradigm. **q**, **t** Representative images of NpHR and ChR2 expression in IL–pBLA neurons. Scale bar = 200 μm. **r**, **s**, **u**, **v** Effects of IL–pBLA manipulation on the RP proportion and the first poking latency. Two-way ANOVA group × epoch interaction; first poking latency [NpHR-RP] F(1,34) = 6.969, $P = 0.0282$, Bonferroni post hoc analysis, \*$P < 0.05$ vs eYFP: IL-pBLA light on, n = 9 (eYFP:IL-pBLA) or 10 (NpHR: IL-pBLA) mice; [ChR2-AP] F(1,46) = 6.085, $P = 0.0163$, Bonferroni post hoc analysis, \*$P < 0.05$ vs eYFP: IL-pBLA light on, n = 10 (eYFP:IL-pBLA) or 15 (ChR2: IL-pBLA) mice. Data were presented as mean ± SEM. Experiments in **b**, **d**, **e i**, **q**, **t** were successfully replicated at least three independent times. Source data are provided as a Source Data file.

intraperitoneally injected into the CUMS mice, the improvement of IL–pBLA inputs on anxiety- and depression-like behaviours was almost abolished, as evidenced by the remarkably reduced centre exploration time in OFT (Fig. 5b–d), decreased open-arm entries and open-arm exploration time with increased open-arm entry latency in EPM (Fig. 5e–g), and a declined sucrose preference in SPT (Fig. 5h). These data suggest that pBLA$^{Calb1}$ neurons are essential in suppression of anxiety- and depression-like behaviours by IL–pBLA pathway.

Together, these data demonstrate that deficits in the IL–pBLA$^{Calb1}$ circuit induce the impairment of reward generalization and susceptibility to anxiety/depression-like behaviours. Excitingly, activating the IL–pBLA$^{Calb1}$ circuit can enhance reward generalization to buffer stress in uncertainty, and consequently ameliorate anxiety- and depression-like behaviours (Fig. 5i).

## Discussion

Natural stimuli rarely appear in exactly the same form from encounter to encounter, therefore, the ability to generalize learning across stimuli and contexts is essential for adaptation. Impaired reward generalization[22] and excessive fear generalization[41] which lead to maladaptation are linked to anxiety-like behaviours. Previous research suggests that mPFC and BLA may play a role in both positive and negative emotions, but whether and how their monosynaptic connection governs reward generalization and modulate anxious states is currently unknown. Here, we developed a new go/go task, by which we were able to understand reward and aversion generalization in response to ambiguous cues. We found that when facing uncertainties, some mice preferred approaching for reward to the ambiguous cue with robust activation of pBLA$^{Calb1}$ neurons. Combining neural circuit tracing and electrophysiology recording, we dissected a monosynaptic excitatory IL–pBLA$^{Calb1}$ connection. Functionally, the IL–pBLA pathway was dramatically activated during the prediction of positive behavioural responses for ambiguity. Stimulating the IL–pBLA pathway, or simply targeting pBLA$^{Calb1}$ neurons, remarkably promoted the reward generalization, while the inhibition produced opposite effect. Reward generalization was negatively related to anxiety- and depression-like behaviours. In CUMS mice, stimulating the IL–pBLA pathway robustly ameliorated anxiety- and depression-like behaviours in a manner dependent of pBLA$^{Calb1}$ neuron activation. Together, this study reveals that a novel IL–pBLA$^{Calb1}$ circuit can drive reward generalization in uncertainties and consequently ameliorate anxiety- and depression-like behaviours.

For uncertainties, humans, and animals usually generalize previously learned information to direct adaptive behaviours[42,43]. We found that mice showed typical poke responses for sucrose reward in response to a novel tone after RC training. This reward-associated approaching behaviour should not be interpreted by inaccurate memory associated with the CS tone, because the poking times were lower and the first poking latency was longer in response to novel tone than those to the accurate CS tone. Furthermore, the HRG and LRG

mice had similar correct poking times in the CS-tone test. Our findings of reward generalization are consistent with Yusuke's work using a head-restrained setup[44]. To further investigate reward generalization in uncertainties, we developed a new go/go paradigm in which mice experienced both reward and aversive conditioning. By presenting an ambiguous probe tone, we were able to understand the preference of reward generalization and aversion generalization inferred from active responses in mice. This paradigm could, at least to certain extent, avoid the interference of response omission to the experimental results in the "no-go" paradigm by which the existence of negative emotion bias was found in unpredictably stressed rats[34]. Excitingly, we could clearly distinguish the more reward-generalized mice from the more aversion-generalized mice by this novel go/go paradigm, although the paradigm may be further improved to reduce the difficulty differences between poking for reward and stepping on platform for aversive avoidance. Woon et al. has previously reported that prior stress could significantly reduce reward seeking behaviours to a reward-associated cue during training[38]. In our go/go paradigm, we found a dramatical reduction of reward-generalized approaching in mice with more aversive generalization, and vice versa. These findings indicate that there may be a mutual inhibition between reward and aversion generalization, and the underlying mechanisms deserve further study.

Along the anterior-posterior axes, the anterior part (aBLA) and posterior part (pBLA) of the basolateral amygdala are recruited by stimuli that elicit negative and positive behaviours[23,37,45], respectively. Targeting aBLA[23,45] and pBLA[23,37,45] governs different affective behaviours. Recently, Burgos-Robles et al. developed a conflicting behaviour paradigm and found that BLA neurons projecting to the prelimbic medial prefrontal cortex (PL) are dramatically implicated in predicting behavioural responses during competition signals[8]. As a site of CS-US convergence, the BLA is also involved in fear[46–51] and fear generalization[52,53], by which it modifies avoidance behaviours in the presence of ambiguous information. In the present study, we focused on the role of pBLA neurons, especially the pBLA$^{Calb1}$ neurons, in the reward generalization. Interestingly, we found that activation of pBLA$^{Calb1}$ neurons was coincident with the reward-generalized approaching for ambiguity. Stimulation of pBLA$^{Calb1}$ neurons enhanced the preference of reward generalization in response to the ambiguous cue. The opposite effects were observed when BLA$^{Calb1}$ neurons were inhibited. Our data not only support the role of pBLA neurons in positive affective behaviours, but also identify their novel function in triggering reward generalization for ambiguity.

The aBLA receives abundant projections from other brain regions, such as the prefrontal cortex (PFC)[54,55], locus coeruleus (LC)[56], anterior cingulate cortex (ACC)[57], cortex[58], thalamic reticular nucleus (TRN)[59], and hippocampus[60,61]. However, the upstream of pBLA neurons, especially pBLA$^{Calb1}$ neurons, are still unknown. In the present study, we identified a novel IL–pBLA$^{Calb1}$ circuit, and found that this circuit was robustly activated with reward-generalized approaching to

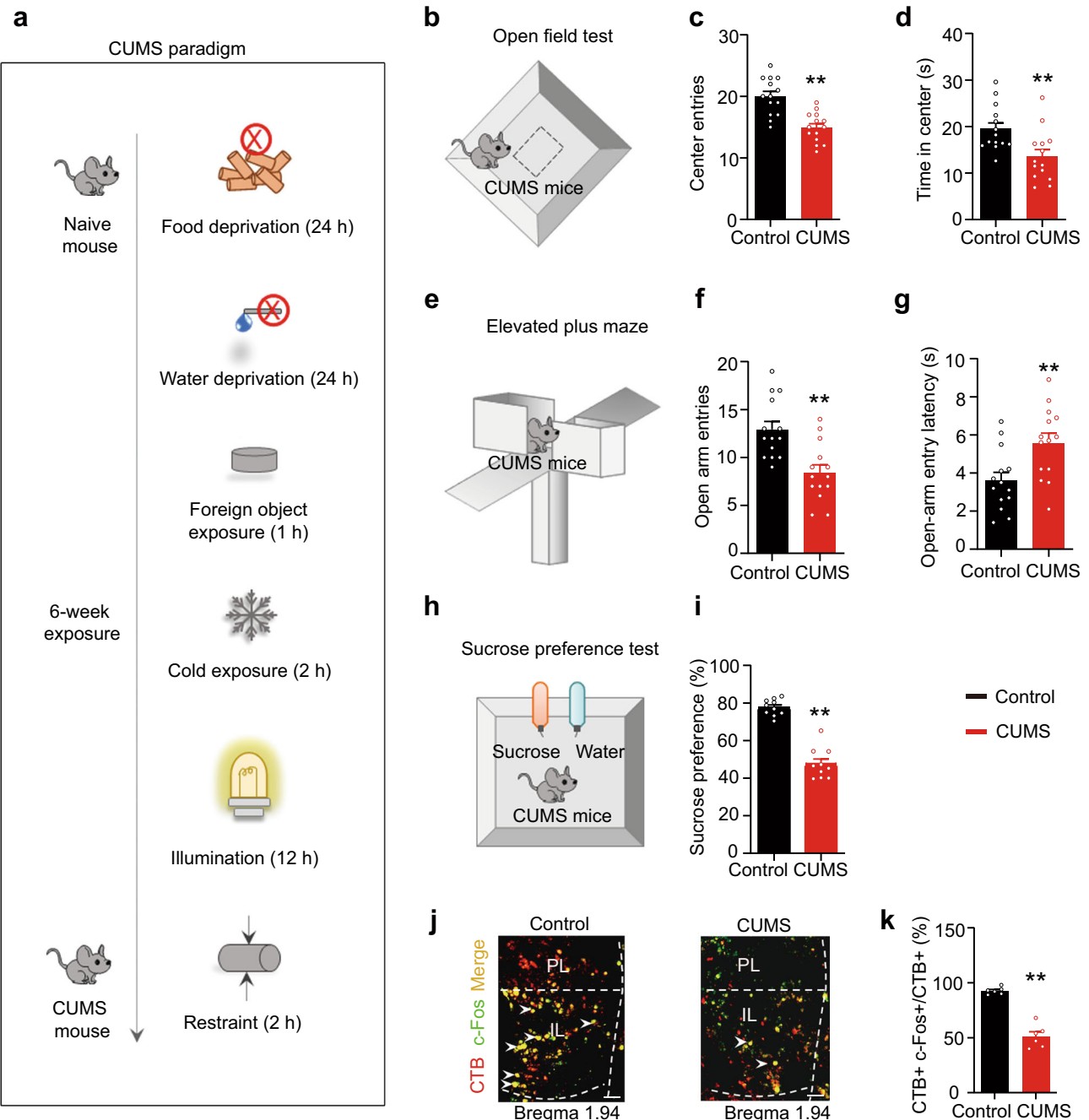

**Fig. 4 | CUMS induces anxiety- and depression-like behaviours with IL–pBLA pathway inhibition. a** Scheduling for Chronic Unpredictable Mild Stress (CUMS). **b**, **e**, **h** Schematic of open field test (OFT, **b**), elevated plus maze (EPM, **e**) and sucrose preference test (SPT, **h**). **c**, **d** Compared with controls, CUMS mice showed fewer centre entries and less time staying in the centre during OFT. $n = 14$ mice per group. Two-sided Unpaired $t$ test, [Centre entries] $t = 4,926$, $df = 26$, $P < 0.0001$; [Time in centre] $t = 2.997$, $df = 26$, $P = 0.0059$. **f**, **g** In EPM, the CUMS mice showed a decreased open arm entry (**f**) and an increased entry latency to the open arm (**g**). $n = 14$ mice per group. Two-sided unpaired $t$ test, [Open arm entries] $t = 3.868$,

$df = 26$, $P = 0.0007$; [Open-arm entry latency] $t = 2.996$, $df = 26$, $P = 0.0059$. **i** In SPT, the CUMS mice showed less sucrose preference than the control mice. Two-sided unpaired $t$ test, $t = 10.30$, $df = 18$, $P < 0.0001$. $n = 10$ mice per group. **j** Representative co-staining of CTB+/c-Fos+ neurons in the IL shown by pBLA injection of CTB (red) and measured at 90 min after EPM. Scale bar = 100 μm. **k** The percentage of both CTB+ and c-Fos+ over CTB+ was significantly decreased in CUMS group than controls. $n = 6$ mice per group. Two-sided unpaired $t$ test, $t = 9.341$, $df = 10$, $P < 0.0001$. **\*\*$P < 0.01$ vs Control. Data were presented as mean ± SEM. Source data are provided as a Source Data file.

the ambiguous cues, indicating a role of the IL–pBLA circuit in the reward generalization. Further, we found that manipulating IL–pBLA circuit controls reward generalization, not aversion generalization, and photostimulating PL–pBLA circuit had no effect on reward preference. These data suggest the specific role of IL–pBLA circuit on the reward generalization. Activation of the IL–pBLA pathway significantly reduced the anxiety- and depression-like behaviours of CUMS mice in a

pBLA[Calb1] neuron activation-dependent manner. In addition to fear extinction[62–65], previous study revealed that IL could determine the stressor controllability and then block the behavioural sequelae of uncontrollable stress[66], exerting anxiolytic[67] and antidepressant[68,69] effects, which were in line with our novel findings of the IL–pBLA[Calb1] circuit. Since the BLA shares reciprocal projections with the IL[27,54,69,70], it will be interesting in future studies to determine whether generalized

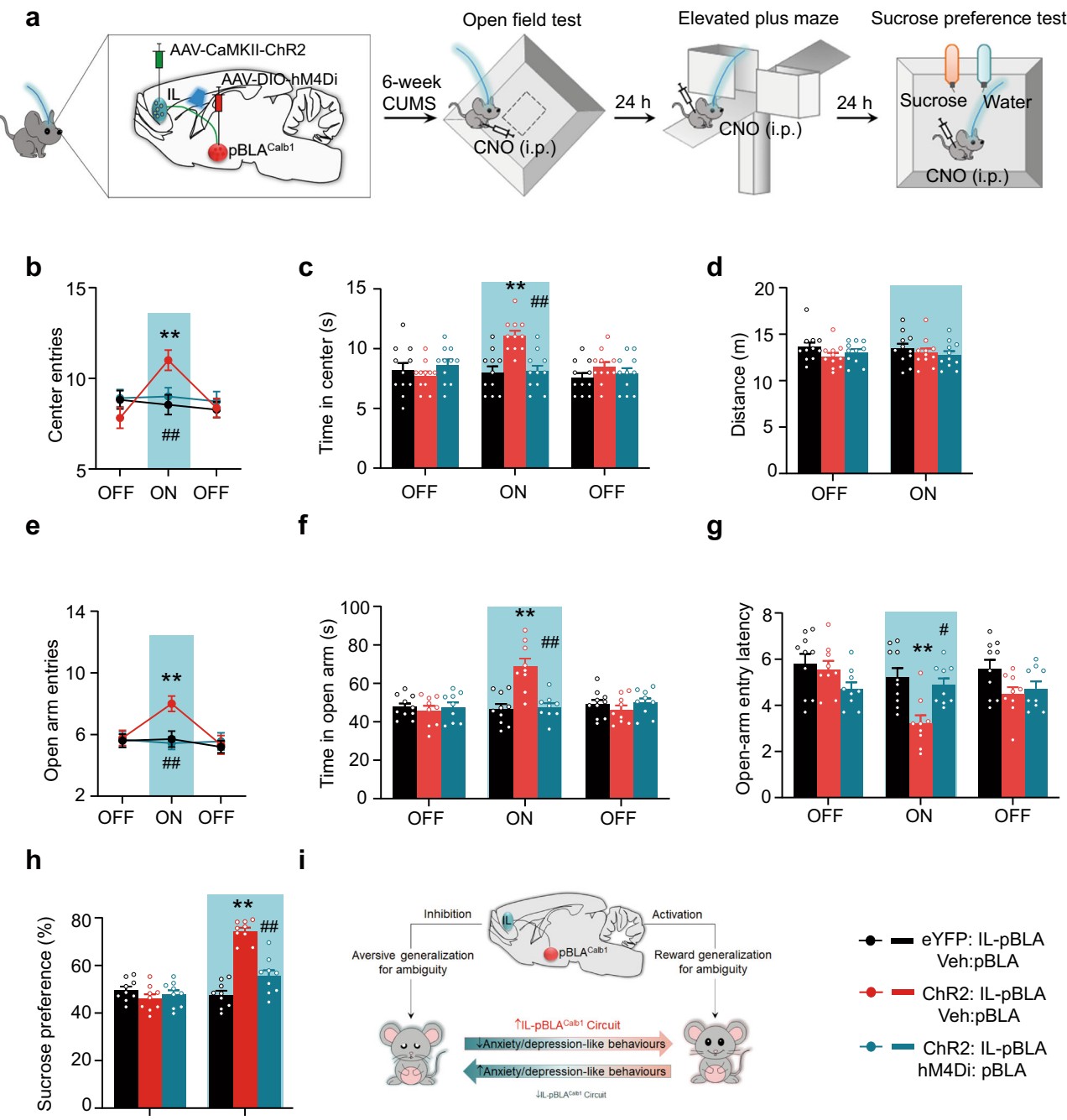

**Fig. 5 | Activating IL–pBLA^Calb1 circuit ameliorates anxiety- and depression-like behaviours in CUMS mice. a** Schematic of IL–pBLA photostimulation and pBLA^Calb1 inhibition in CUMS mice during OFT, EPM, and SPT. AAV-CaMKII-ChR2 and AAV-DIO-hM4Di were injected into the IL and pBLA of Calb1-Cre mice, respectively. At the 5th week of CUMS training, fibres were implanted in the pBLA. 30-32 min before behaviour tests, CNO was intraperitoneally injected. **b, c, e–g, h** ChR2-CUMS mice were tested in a single 9 min session in OFT or EPM with three 3 min epochs. ChR2-CUMS mice showed less anxiety- and depression-like behaviours during the illumination epoch, evidenced by greater centre entries (**b**) and time staying in the centre (**c**) in OFT, more open-arm exploration (**e, f**) and shorter open-arm entry latency (**g**) in EPM, and higher sucrose preference in SPT (**h**), while the suppression on anxiety- and depression-like behaviours were reversed when Calb1+ neurons in the pBLA were inhibited by CNO injection. No difference on travelled distance was detected during and after light stimulation or CNO injection (**d**). Two-way ANOVA

group × epoch interaction. In OFT: $n = 11$ mice per group, [Centre entries]: $F(4,90) = 3.611$, $P = 0.0089$; [Time in centre]: $F(4,90) = 5.541$, $P = 0.0005$; [Distance moved]: $F(2,60) = 0.3377$, $P = 0.7174$. In EPM: $n = 9$ (ChR2:IL-pBLA, HM4D: pBLA +ChR2:IL-pBLA) or 10 (eYFP:IL-pBLA) mice, [Open arm entries]: $F(4,75) = 2.651$, $P = 0.0396$; [Time in open arm]: $F(4,75) = 9.704$, $P < 0.0001$; [open-arm entry latency]: $F(4,75) = 3.344$, $P = 0.0142$. [Sucrose preference]: $n = 9$ mice per group, $F(2,48) = 34.86$, $P < 0.0001$. Bonferroni post hoc analysis, **$P < 0.01$ vs eYFP:IL-pBLA +veh:pBLA; #$P < 0.05$, ##$P < 0.01$ vs ChR2:IL-pBLA+veh:pBLA. Data were presented as mean ± SEM. **i** Proposed working model. The excitatory IL–pBLA^Calb1 circuit drives reward generalization for ambiguous information in a reversible manner. Activating IL–pBLA^Calb1 pathway promotes reward generalization for the ambiguity, reducing anxiety- and depression-like behaviours. Source data are provided as a Source Data file.

information could be conveyed back to the IL for ultimate integration within the IL–pBLA loop circuit.

In summary, we identified a novel IL–pBLA^Calb1 circuit and decoded its crucial role in reward generalization for the ambiguity. Importantly, the IL–pBLA^Calb1 circuit may serve as a promising target to buffer stress, and thus ameliorate anxiety- and depression-like behaviours.

## Methods

**Animals.** Adult male C57BL/6 mice (6–8 weeks old, Supplementary Table 1) were obtained from Beijing Vital River Laboratory Animal Technology Co., Ltd. Calb1-IRES2-Cre-D mice and tdTomato reporter Ai9 mice were kind gifts from Prof. Xiaohui Zhang (State Key Laboratory of Cognitive Neuroscience & Learning and IDG/McGovern Institute for Brain Research, Beijing Normal University). Except for the CUMS model, the animals were housed in groups with three to five per cage and were kept on a 12 h light to 12 h dark cycle (lights on at 6:00 p.m., off at 6:00 a.m.) and were settled at a stable temperature (23–25 °C) and consistent humidity (50 ± 5%). The animals had free access to food and water. All animal experiments were performed according to the "Policies on the Use of Animals and Humans in Neuroscience Research" revised and approved by the Society for Neuroscience in 1995, and the Guidelines for the Care and Use of Laboratory Animals by the Ministry of Science and Technology of the People's Republic of China. The Animal Care and Use Committee at Tongji Medical College, Huazhong University of Science and Technology approved the study protocol.

### Anterograde tracing
Mice were anaesthetized using 1.5% isoflurane at an oxygen flow rate of 1 L/min and the head was fixed in a stereotactic frame (Stereotaxic for Mouse, SGL M,68030 Adaptor Inci; RWD Life Science Co., Ltd.). For all surgeries, body temperature was maintained with a heat lamp and eyes were coated with an erythromycin ointment. For anterograde poly-synaptic tracing, anterograde HSV (100 nl, $2–5 \times 10^9$ pfu/ml, Supplementary Table 1) was purchased from Brain VTA Technology Co., Ltd. The virus was injected into the IL (1.9 AP, ±0.3 ML, and −3.0 DV). 36 h later, mice were treated with a lethal dose of sodium pentobarbital via intraperitoneal injection, and then the mice were transcardially perfused with ice-cold 0.9% saline followed by 4% paraformaldehyde. The expression of GFP in the posterior basolateral amygdala was observed by a confocal microscope (LSM 780, Zeiss, Germany).

### CTB retrograde tracing
Recombinant cholera toxin subunit B (CTB, 100 nl, Brain VTA Technology Co., Ltd., Supplementary Table 1) was delivered stereotaxically into the pBLA (−2.3 AP, ±3.4 ML, and −4.85 DV). Seven days later, the mice were anaesthetized and perfused as mentioned above. The signal of CTB in the IL was observed with a confocal microscope (LSM 780, Zeiss, Germany).

### Patch-clamp electrophysiology
The brains were sliced in ice-cold artificial cerebrospinal fluid (ACSF) containing 124 mM NaCl, 3.0 mM KCl, 1.0 mM MgCl₂, 2.0 mM CaCl₂, 1.25 mM NaH₂PO₄, 26 mM NaHCO₃, and 10 mM glucose and saturated with 5% CO₂ and 95% O₂ (pH = 7.4). Then, the slices were incubated for 30 min at 32 °C in the same solution and equilibrated to room temperature for at least 30 min. Whole-cell patch-clamp recording was carried out by visually identified pBLA^Calb1+ neurons (tdTomato+ neurons) in Calb1-IRES2-Cre-D::Ai9 mice after 4–5 weeks of AAV-CaMKIIa-hChR2(H134R)-eYFP infection in their IL. All recordings were performed by a Multiclamp 700B amplifier (Molecular Devices, Sunnyvale, CA). Analogue signals were first low-pass filtered at 1 kHz and then digitized at 10 kHz by Digidata 1440 and pClamp10 software (Molecular Devices, Sunnyvale, CA). ACSF and drugs were used to the slice by a peristaltic pump (Minipuls3; Gilson, Middleton, WI) at 2 ml/

min. To activate IL–pBLA terminals, square pulses of blue light (472 nm, duration of 5 ms) were transferred through a ×40 water-immersion objective. LED served as a light source. The light power at the microscope objectives was ~2 mW/mm². Off-line analysis was performed by Clampfit 10 (Molecular Devices, Sunnyvale, CA). For validation of CNO effects on hM3Dq and hM4Di, 5 μM CNO in ACSF was applied via bath application after at least 10 recordings of stable action potential firing.

### Optogenetic manipulation in free-moving mice
AAV-CaMKIIa-hChR2(H134R)-eYFP (Supplementary Table 1) and AAV8-CaMKIIa-eNpHR3.0-eYFP (Supplementary Table 1) were purchased from Obio Technology (Shanghai, China). Virus (100 nl, >10^12 vg/ml) was injected into the IL (1.9 AP, ±0.3 ML, and −3.0 DV) at a speed of 0.1 μl/min. Four to five weeks later, optical fibres (core = 200 μm; numerical aperture = 0.37) were implanted in the pBLA (AP: −2.3 mm, ML: ±3.45 mm, DV: −4.35 mm). After 1 week of recovery, the behavioural tests were performed by optogenetically manipulating the IL–pBLA circuit in free-moving mice. For Calb1 neuron manipulation, AAV8-CaMKIIa-DIO-hChR2(H134R)-eYFP (Supplementary Table 1) and AAV8-CaMKIIa-DIO-eNpHR3.0-eYFP (100 nl, >10^12 vg/ml, Supplementary Table 1) were injected into the pBLA (AP: −2.3 mm, ML: ±3.45 mm, DV: −4.85 mm) of Calb1-Cre mice. Optical fibres (core = 200 μm; numerical aperture = 0.37) were implanted in the pBLA.

### Chemogenetic manipulation in free-moving mice
AAV8-CaMKIIa-hM3D(Gq)-eYFP (Supplementary Table 1), AAV8-CaMKIIa-hM4D(Gi)-eYFP (Supplementary Table 1), AAV8-CaMKIIa-DIO-eYFP (Supplementary Table 1) and AAV8-CaMKIIa-DIO-hM4D(Gi)-eYFP (Supplementary Table 1) purchased from Obio Technology (Shanghai, China), was injected (100 nl, >10^12 vg/ml each) into the pBLA (AP: −2.3 mm, ML: ±3.45 mm, DV: −4.85 mm) at a speed of 0.1 μl/min. After 4 weeks, the expression of the virus was approved. The hM4D (Gi) or hM3D (Gq) was activated with clozapine-N-oxide (CNO, Sigma-Aldrich, Supplementary Table 1) dissolved in 0.9% saline containing 5% DMSO. Intraperitoneal injection of CNO was conducted 30–32 min before behavioural testing at a dose of 10 mg/kg to inhibit or activate neuronal activity.

### Immunostaining
Mice were sacrificed 90 min after the last trial of the behavioural tasks by a lethal dose of sodium pentobarbital. The brains were removed after transcardial perfusion, and then post-fixed overnight using 4% paraformaldehyde (PFA), after which they were put into 25–30% sucrose solution for 3 days in PBS and then sliced in 30 μm thick coronal sections. For immunofluorescence, sections were washed in PBS-T (PBS consists of 0.1% Triton X-100) and subsequently incubated using Calbindin (1:500; Cat no. Ab108404, Abcam), Calbindin D-28k (1:500; Code No:300, Swant) and/or c-Fos (1:300; Cat no. 226003, Synaptic Systems) for 17–20 h. Followed by 1 h incubation with the secondary antibody using Alexa Fluor 488 donkey to rabbit IgG(H+L), 1:300, Invitrogen A21206; using Alexa Fluor 488 donkey to mouse IgG(H+L), 1:300; Invitrogen A21202 or Alexa Fluor 546 Donkey Anti-Rabbit IgG, 1:300, Invitrogen A10040 at 37 °C, the sections were washed with PBS-T (10 min, three times). Finally, the slices were counterstained with DAPI. Fluorescence images were analyzed by Image J 64. Information of antibodies for immunostaining was listed in Supplementary Table 2.

### Skinner box behavioural tasks (go/go task)
Mice were trained in standard operant chambers (22 cm × 18 cm × 18 cm; length × width × height, Wuhan Yihong Technology Co., Ltd), which were located in sound-attenuating cubicles. Each chamber was equipped with a house light and speakers for the delivery of tone. A syringe pump was used to deliver sucrose, a sucrose port was

equipped with an infrared beam for the detection of entries and exits, and a grid floor was used for the delivery of electrical shocks. After testing each animal, chambers were wiped with 75% ethanol. All training phases occurred in this context.

The first phase of training included the acquisition of a Pavlovian reward association, in which mice learned to use sucrose to associate a conditioned stimulus (i.e., CS-Suc). To promote reward acquisition, mice were pre-exposed to sucrose multiple times in the home cage and in the training chambers. Reward conditioning included the presentation of a positive tone (2000 Hz or 9000 Hz, 75 dB) that lasted for 30 s and predicted the delivery of a 30% sucrose solution[8] (5 μL per trial). Sucrose was delivered for 30 s during the cue presentation. Mice underwent 16-day training (~35 trials per day, 4 days per session). In four reward sessions, the inter-trial intervals (ITI) were variable, lasting from 10 s, 20 s, 30 s to 50 s. After cue offset, sucrose was removed by vacuum immediately if mice did not obtain it during the CS. On the last day of every session, the training went on until the animals reached a minimum of 70% successful responses to positive poke following the positive tone presentations.

For CS-social reward training, an automated operant conditioning system consisting of two chambers (a smaller chamber to hold the target animal and a larger chamber to place the subject animal) was used (Supplementary Fig. 3a). The two chambers were separated by a metal wire grid (5 cm × 10 cm) to prevent the animal from going to the other chamber but to allow the subject animal to sniff and research the target animal which is an unfamiliar juvenile male. An opaque plastic door located behind the wire grid can be opened or closed by a motor. Beside the door, there was a nose-poke port that could be detected by infrared beam sensors. Similar to CS-Suc, the mice received social reward training by exposure to the positive tone (2000 Hz or 9000 Hz, 75 db) for 30 s each trial, in which the nose-poking the port initiated the opening of the door followed by the social interaction for 7 s; the mice underwent a total of 16 days of training, 35 trials per day, 4 days per session with ITIs of 10 s, 20 s, 30 s and 50 s at each session. On the last day of every session, the training went on until the animals reached a minimum of 70% successful responses to positive poke following the positive tone presentations.

The second phase of training included the acquisition of association between aversive shocks and conditioned stimulus (i.e., CS-Shock). There was an insulating rubber platform (3.0 cm high and 5.0 cm wide) placed on the floor. When the negative tone (9000 Hz or 2000 Hz, 75 dB) was on, the aversive shocks (10 s duration, 0.25 mA current intensity) were delivered and then terminated as soon as the animals climbed on the platform. At the beginning, the animals generally stepped down after 2–10 s of staying on the platform, and then the foot shocks were delivered again. When the animals were trained to hold on to the platform or stay on the platform with at least three limbs for up to 30 s, we stopped training for the trial. The procedure was repeated for 7 consecutive days, with 10 trials per day except 10–20 trials on the first day. Considering the variation in the step-down latency of each mouse during each training trial, we changed the ITI randomly from 45 to 60 s among 10 trials of each day. Latency to step on the platform and staying time on the platform were recorded. These positive and negative conditioning stimulation tones were counterbalanced across animals.

Then, we conducted RC and FC tests in which 10 positive tones or 10 negative tones were presented and separated by 50 s ITIs. A trial was recognized to be correct when an animal carried out the response associated with each tone as learned in CS-Suc and CS-Shock training. No response or wrong response was considered as a failure trial. To avoid the remedial poking behaviour as shown in classic platform-mediated avoidance[35,36], we manually switched off the sucrose delivery system if the animal chose to step on the platform in response to the positive CS tone. Therefore, it was impossible for the mice to use the ITI to poke for the sucrose reward. If the animal briefly touched the

platform (less than three limbs) and quickly turned back to poke for sucrose (turn-back latency is always less than 5 s), the trial should be labelled as a false response. Training was continued until animals reached a stable baseline of correct discrimination responses on at least 70% of the trials.

In the ambiguous-cue examination phase, 10 intermediate tone (5000 Hz, 75 dB) trials were presented and mice were examined for their corresponding responses to the ambiguous tones. In each trial, an ambiguous tone was stopped immediately after the mice poked or step on the platform.

### Chronic unpredictable mild stress (CUMS)

The mice were transferred to a clean room and then various unpredictable chronic mild stressors were sequentially delivered for 6 weeks as follows: food deprivation for 24 h, water deprivation for 24 h, foreign object exposure for 1 h, night illumination for 12 h, cold exposure for 2 h, and physical restraint for 2 h. The procedures were repeated from week 1 after the first 3 weeks. Some stressors were interrupted or modified when they occurred at particular points in the sucrose-preference test schedule (Supplementary Table 3). Nonstressed mice were left undisturbed in their own cages except during housekeeping procedures.

### Elevated plus maze (EPM) test

Mice were placed in a standard EPM sized maze, which was elevated 50 cm from the floor and included two closed arms (66 cm × 6 cm), and two open arms (66 cm × 6 cm), with a central area (6 cm × 6 cm). Mice underwent a 5 min test (Fig. 4) or performed in a single 9-min test with three 3 min epochs (Fig. 5). The test began with a light-off baseline epoch, then a light-on illumination epoch followed, and finally followed by a second off epoch. During the light-on period, a constant 20 Hz blue light (5 ms pulses, 5–8 mW, 472 nm) or yellow light (8–10 mW, 589 nm) was delivered onto IL−pBLA terminals or pBLA Calb1+ neurons. The number of entries into the open arm and the time the mice stayed in the open arm were recorded. The maze was scrubbed with 75% ethanol between each trial.

### Open field test (OFT)

A transparent plastic chamber (50 cm × 50 cm × 40 cm, length × width × height) was used as the open field and divided into a central zone (25 cm × 25 cm) and a peripheral zone. Mice were placed in the centre of the chamber before starting the test. The test lasting 5 min (Fig. 4) or 9 min (Fig. 5) was the same as that in EPM. The centre entries, the distance moved per min and the time spent in the central area were recorded by video-tracking and behavioural analysis software (Chengdu Techman Software Co., Ltd).

### Sucrose-preference test (SPT)

Seventy-two hours before the test, mice were trained to adapt to 1% sucrose solution (w/v): two bottles of 1% sucrose solution were placed in each cage, and 24 h later, 1% sucrose in one bottle was replaced with tap water for 24 h. After the adaptation, mice were deprived of water and food for 24 h. A sucrose preference test was conducted during the daily activity period of the mice in which mice were housed in individual cages and were free to access to two bottles containing 100 ml of sucrose solution (1% w/v) and 100 ml of water, respectively. After 1 h, the volumes of consumed sucrose solution and water were recorded and the sucrose preference was calculated by the following formula: sucrose preference = sucrose consumption/ (water consumption + sucrose consumption) × 100%.

### Fibre photometry in the skinner box behavioural tasks

The change in neuronal activity in the behavioural tasks can be examined by recording GCaMP6 fluorescent signals with the optical fibre recording system (Thinker Tech Nanjing Biotech Limited Co.,

Ltd). Four weeks after the injection of AAV-CaMKIIa-DIO-GCaMP6f virus (100 nl, >10$^{12}$ vg/ml each) into the IL, an optical ceramic needle was inserted into the IL through the craniotomy. Using an optical fibre recording system, a 488 nm laser (0.01–0.02 mW) was delivered, and that fluorescent signals were recorded. By data analysis, the original signal was demodulated and converted to df/f. Normalized df/f could monitor the activity alterations in the IL–pBLA connection. Manual tagging and motion tracking were used to mark the position and monitor the activity of mice when they were experiencing skinner box behavioural tasks. Data were analyzed using MATLAB 2017a.

### Statistical analyses

Commercial software (GraphPad Prism version 8; GraphPad Software, Inc, La Jolla, CA) was used for statistical analysis, via $t$ tests, one-way ANOVAs and two-way ANOVAs to define the different means among the groups. The normality of the data distribution was confirmed by the D'Agostino-Pearson omnibus normality test and using the F test, Brown–Forsythe test and Bartlett's test, all the data fit homogeneity of variance. The data were shown as mean ± SEM and the significance threshold was set at $P = 0.05$. All data generated in this study are provided in the Source Data file.

### Reporting summary

Further information on research design is available in the Nature Research Reporting Summary linked to this article.

## Data availability

The data supporting the findings of this study are available within the article and Supplementary Information files. All the source data generated in this study are provided in the Source Data file. Source data are provided with this paper.

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

## Acknowledgements

This study was supported by grants from the Natural Science Foundation of China (No. 82071219 to Y.Y., and Nos. 91949205, 31730035, and 81721005 to J.Z.W.), Science and Technology Committee of China (No. 2016YFC1305800 to J.Z.W.), the Special Project of Technological Innovation of Hubei Province (No. 2018ACA142 to J.Z.W.), and Guangdong Provincial Key S&T Programme (No. 018B030336001 to J.Z.W.).

## Author contributions

This study was initiated and designed by J.Z.W. and Y.Y. J.Z.W. directed and coordinated the study. H.Y. performed major animal behaviour studies, virus injections and tracing, optogenetic, chemogenetic manipulations. L.C. recorded the population calcium signalling and performed imaging analysis. H.L. performed brain slice electrophysiology recordings. G.P., W.P., B.H., G.S., and Y.G. performed part animal behaviour studies. R.X., J.T., and F.S. performed part of immunohistochemistry. G.P., H.L., Z.J., and D.W. performed genotyping. L.C. and G.P. performed part of the optogenetic manipulations. L.C., X.W., G.P., Y.L., and J.L. analyzed the behaviour data in double-blind way; H.Y., G.P., and Y.L. analyzed the imaging data; D.K. helped to collect the data; Y.Y. and J.Z.W. analyzed the data and wrote the manuscript.

## Competing interests

The authors declare no competing interests.

## Ethical approval

The procedure was approved by the Animal Protection and Use Committee of Huazhong University of Science & Technology. This manuscript does not contain patient data or clinical studies.
