## [Peer Review File · Nature Communications]

Infralimbic medial prefrontal cortex signalling to calbindin 1 positive neurons in posterior basolateral amygdala suppresses anxiety- and depression-like behavioursREVIEWER COMMENTS

Reviewer #1 (Remarks to the Author):

The authors reports several experiments undertaken in mice to identify the neural underpinnings of optimistic or pessimistic behaviour, by using several methods, including behaviour, chemogenetic, optogenetic anterograde and retrograde tracing, immunohistochemistry. It is focusing on the projection from the infralimbic cortex to the posterior part of the Basolateral amygdala. This topic is interesting, as few experiments have tried to identify the neural basis of optimism or pessimist, and as such it adds really new findings to the literature.

Here are my concerns:

- a) Regarding the Pavlovian reward association, a sucrose solution at 30% is used we seem really very sweet. The choice of this should be further justified. It would also be interesting to check whether other using other kind of rewards, the reward conditioning is still effective?
- b) in the introduction, it would be important to explain the rationale for investigate the role of the infralimbic cortex. indeed, the introduction provides argumentation to focus on the posterior BLA, but o rationale for the infralimbic cortex is provided in this section
- c) do the MPI and LPI subgroups display differential sensitivity to CUMS?
- d) in all experiments using optogenetics or chemogenetics, it is mandatory to provide data showing the optogenetic/chemogenetic excitation or silencing actually changed the activity pattern of the targeted area or projection. That is: does excitation of the pBLA really exhite pBLA cells and similarly does inhibition of the pBLA inhibit pBLA cells? and so on for all experiments presented. Without this control, authors cannot claim that actually they did excite or inhibit the areas or the projections. This is an important concern. This is necessary to reach the standard of the field.
- e) in the same vein as concern d), in all experimnets using viral injections (optogenetics, chemogenetics), figures showing the area of the AAV expression 4 weeks after the injection should be provided, showing its extension in the targeted area, as well as non expression in adjacent areas. This is necessary to reach the standard of the field.
- f) did data fit normality of distribution and homogeneity of variance? it not, Anovas cannot be used
- g) the CUMS protocol includes food or water deprivation lasting 24 hr. This will interfere with sucrose drinking and with exploratory behaviour. So a table should be provide showing a timeline with the stressors, and the moment of the the behaviour testing, in order to check absence of interference.

Reviewer #2 (Remarks to the Author):

Yu and colleagues have used a series of behavioral, recording, and projection-specific manipulation studies to examine the role of IL->BLA calbindin cells in modulating behavioral responses related to reward and aversion. While the use of several advanced tools in combination is impressive, the overall rationale for the studies and interpretations presented are problematic for several reasons.

The premise of the model used, as well as interpretations throughout, are overly anthropomorphic. The paper is framed around an idea that we can understand cognitive attributions of mice as being optimistic or pessimistic, and further, it attempts to link anxiety and depression to the findings presented. This is entirely inappropriate for studies using

rodents. The language of the paper and framing of the data need to be completely overhauled.

Second, what's described as 'optimistic or pessimistic preference' seems to actually be basic generalization and seemingly random/guessing behavior to an ambiguous cue. There have been many studies in the field on cue generalization and ambiguity for both appetitive and aversive learning, as well as studies combining appetitive and aversive cues in the same task. None of these are cited or mentioned. Rather, the simple phenomenon of generalization is framed as an inherent mouse cognitive bias - a much less parsimonious explanation for the observed behavior. Before the data here can be properly interpreted, I suggest the authors compare their findings to these related studies in the field and present them using the language already established in the field.

Other minor issues:

- In the RC task, mice are defined as more/less positive interpretation (MPI or LPI) depending on whether they respond to the ambiguous cue. MPI mice are simply mice that generalize more than LPI mice.

- With regard to negative preference (NP) and positive preference (PP) mice, with a ratio of NP:PP close to 50:50, this seems to further imply that what's going on here is random sampling/choice.

Reviewer #3 (Remarks to the Author):

In this manuscript, the authors investigate the role of the posterior basolateral amygdala Calbindin 1 expressing neurons (pBLACalb1), and the infralimbic (IL) - pBLACalb1 circuit in positive interpretation bias of an ambiguous cue in a go/go task that the group developed. The authors first use cFos staining to identify increased activity in IL and pBLACalb1 neurons following a short-latency nose poke for reward to a novel cue after initial reward conditioning to a different cue. The authors then show that cFos increases in the pBLACalb1 in a go/go paradigm, after a positive response (nose poke as for a reward cue rather than platform avoidance as for an aversive cue) to a novel cue, after training on both reward and fear conditioning with two different cues. The authors then show that optogenetic excitation and inhibition of IL input to pBLA bidirectionally increases and decreases, respectively, the positive response rate and latency (nose poke) to a novel cue in the go/go task. Finally, the authors show that after chronic unpredictable mild stress, optogenetic activation of the IL input to the pBLA ameliorates anxiety- and depression-like behaviors in a BLACalb1 – dependent manner. The authors conclude that anxiety and depression like symptoms can be ameliorated by activation of the IL-pBLA Calb1 circuit, thereby suggesting that this pathway is vital for biasing an animal towards optimistic behaviors.

Overall, this manuscript presents interesting data that develop ideas regarding conflicting cue interpretation. Furthermore, the authors develop an interesting go/go paradigm to investigate these questions. However, the manuscript suffers from substantial organizational and grammatical problems that interfere with its ability to fully develop the data. In addition, I have some concerns, described below, regarding the controls used in this study, and the claims that the authors make regarding pessimism and optimism (although this is perhaps a reflection of the language difficulties described above).

Major

- Related to the issue of language, there is misuse of tense, plurality and irregular sentence structure that appear throughout the paper (e.g. paragraph 1 of the introduction, etc). In addition there is quite a bit of anthropomorphizing in the text (e.g. "BLA neurons with their associated circuits interpret...cues", etc.).
- The behavioral data are not sufficiently described to give an understanding of what is being studied (e.g. optimism or pessimism, or positive preference vs negative preference). For example, the reward and aversive nature of the cues aren't established as equally salient for the animals. The 30-s reward cue was paired with sucrose over 16 days of training (with 4 different ITI trained for 4 days each), whereby nose poke to the cue and consumption of reward terminated the trial. (As a side note, the authors show data from a recall test that is described as occurring 24 hrs after training, but it occurs after the animals have been learning and recalling this training for weeks). However, from the description given in the methods, it seems that the fear conditioning employed a different approach, whereby animals were shocked for 10 sec at a time during the duration of the aversive cue (0.25mA), unless they stepped on the avoidance platform. The only behavioral analysis of the fear conditioning paradigm shown is in fig 1c, where Times of Step-On across 7 days of training is shown. Is this the average number of times that the animal stepped onto the platform during trials on a training day? According to the methods, the fear conditioning sessions would be different lengths on different days, because they would stop after animals stayed on the platform for at least 30 seconds. Was the sucrose also available at this time as an incentive to step off the platform to go back to the nose poke? If so, did the mice use the ITI to press for sucrose? Were the animals also showing defensive freezing? How long of each trial did the animals spend on the platform? It would be helpful to analyze these data to reflect training conditions (e.g. number of trials per day, latency to step on the platform), and to help understand the behavior in the go-go paradigm. There is a body of work on fear conditioning and platform avoidance training from Greg Quirk's lab, that this manuscript doesn't make contact with, but which would be helpful for interpretation.
- In several instances it is not clear what controls are being used. For example, the reward conditioning experiment is described in the text as having a no-CS group (line 89). This suggests that there are 2 groups of animals, one that receive CS pairing with reward, and one that was in the box but didn't receive a CS. However, figure 1b shows behavior during the CS+ (black squares) and Pre-CS behavior (grey squares), suggesting that there is 1 group of animals and behavior was scored during the CS, and pre-CS behavior was used as the control condition. Figure 1c refers to a CS- (grey) and a CS+ (black), suggesting that there was a CS that wasn't paired with anything and a CS that was paired with reward. The description in the methods section does not provide any further information.
- For the fiber photometry experiment shown in figure 3, the authors only analyze data from positive preferring mice without controlling for data for negative preferring mice. Did the negative preferring mice show similar changes, or was the IL-pBLA circuit less active in those animals?
- similar to the point above, in experiments shown in figure 2, the IL input to the pBLA was inhibited only in the positive preferring mice, and stimulated only in the negative preferring mice. Did inhibition in negative preferring mice and stimulation in positive preferring mice not affect behavior?
- In several instances the authors use unpaired t-test statistics where rm-ANOVAs, one-way ANOVAs, or two-way ANOVAs would be more appropriate (e.g. fig 1e, f, 1h, etc).
- In figure 2k, the authors show that in positive preferring (nose-poking) mice in the go/go task, the Calb1+ cells make up a larger percentage from the total active pool (Calb1:cfos/cfos) over the negative preferring mice. Does the percent of active Calb1 cells also increase (Calb1:cfos/Calb1)?

-The introduction presents pieces of information (such as the use of a go/go paradigm) without providing prior justification or explanation as to what it is. Although these measures are explained within the methods section, it would benefit the reader to have a lead in as to why they are the appropriate tasks prior to them being introduced as part of the study design.

--The introduction makes numerous claims (e.g. regarding the benefit of optimism and the risks of pessimism), but employs citations sporadically to support these claims (e.g. line 58, "Targeting these BLA neurons remarkably supports positive reinforcement"). Without citations, it is unclear which of these ideas are generated from previous research and which are being concluded by the authors.

Other:

-Placement images throughout the manuscript don't show anatomical landmarks to help identify the outlined regions, making it difficult to fully understand location. For example, figure 1h, figures 3b,d,i, m,n; Figure 4i, Supp 1, etc.

- Figure 1h shows examples of cFos expression in the pBLA that was quantified in 1i. However, I don't see any cFos+ cells in figure 1h. Please show a more zoomed in example of the cells.

-The simplicity of the visual schematics (e.g. fig 5 a, e, i) relative to the complexity of the combined optogenetic/chemogenetic approaches does not capture nuance of the approach.

- Please include a schematic of the CUMS paradigm/ control in the main figures

- Please include an overview of placements across subjects for all experiments.

Reviewer #1 (Remarks to the Author):

The authors reports several experiments undertaken in mice to identify the neural underpinnings of optimistic or pessimistic behaviour, by using several methods, including behaviour, chemogenetic, optogenetic anterograde and retrograde tracing, immuno-histochemistry. It is focusing on the projection from the infralimbic cortex to the posterior part of the Basolateral amygdala. This topic is interesting, as few experiments have tried to identify the neural basis of optimism or pessimist, and as such it adds really new findings to the literature.

[Response]: We greatly appreciate the reviewer for the very positive and encouraging comments to our paper.

Here are my concerns:

a) Regarding the Pavlovian reward association, a sucrose solution at 30% is used we seem really very sweet. The choice of this should be further justified. It would also be interesting to check whether other using other kind of rewards, the reward conditioning is still effective?

[Response]: Thanks to the reviewer for the question. Indeed, sucrose has been widely used in Pavlovian reward association^{1, 2}. Many studies in the field of depression suggest that 1–2% (wt/vol) sucrose solution is the optimal concentration to evaluate anhedonia phenotype in rodents by sucrose preference test³⁻⁷. However, sweeter solution could be optional in cue-directed reward-seeking behaviors. For instance, by using 20% sucrose as a reward, Stuber et al. found that basolateral amygdala to nucleus accumbens excitatory transmission facilitated reward seeking⁸. We had also tried 20% sucrose in our preliminary experiments, but failed to induce a sufficient motivation in our go-go paradigm (please see the original data below). At fourth day of session 4 during FC training (ITI=50 s), %CS-triggered poking was about 69%, which was statistically lower than 30% sucrose group shown in Figure 1 (and the panel b in Figure below). More importantly, the response omission (no poking

response during CS) was much greater in 20% than 30% (panel c in Figure below). These data indicate a sharp loss of reward sensitivity to 20% sucrose after multiple training. Therefore, we used 30% sucrose by referring Burgos-Robles's work⁹. We found excitingly that 30% sucrose was motivational enough to promote poking response when ITI was prolonged to 50 s during FC training (Figure 1b, Figure 2b). We have added the reference for using 30% sucrose in the revised method (please see page 19, line 13).

The preliminary data of 20% sucrose reward conditioning (RC) training and test.

The mice were trained to associate a CS tone (2000 Hz, lasting 30 s) with 20% sucrose reward. (a) The mice showed improved association between CS tones and predicted sucrose. The mice were trained ~35 trials a day, and the intertrial intervals (ITIs) were changed every 4 days (from 10 s, 20 s, 30 s to 50 s) to form 4 different training sessions. The percentage of poking times per day were calculated. $n = 20$ mice per group. Repeated measures two-way ANOVA, [Session 1]: $F(3, 57) = 119.1$, $P < 0.0001$; [Session 2]: $F(3, 57) = 33.36$, $P < 0.0001$; [Session 3]: $F(3, 57) = 8.60$, $P < 0.0001$; [Session 4]: $F(3, 57) = 6.79$, $P = 0.0005$, Tukey's multiple comparisons test, $**P < 0.01$ vs Pre CS (ITI). (b) At fourth day of session 4 during FC training (ITI=50 s), %CS-triggered poking in 20% sucrose group was significantly lower than the 30% sucrose group. Unpaired t test, $t = 9.308$, $df = 38$, $P < 0.0001$. (c) The response omission (no poking response during CS) was much greater in 20% than 30% at fourth day of session 4 during FC training. Unpaired t test, $t = 3.223$, $df = 38$,

$P < 0.0026$. $**P < 0.01$ vs 20% Suc group. Data were presented as mean \pm SEM.

In addition to food reward, social interaction is another kind of reward^{10, 11}. As suggested by the reviewer, we also did additional experiments by using social reward instead of sucrose reward in go-go paradigm (sFigure 3a). After RC (sFigure 3b) and FC (sFigure 3c, d) training, mice successfully established CS-triggered step-on response in FC test (sFigure 3e) and CS-induced social behavior in RC test (sFigure 3f). When an intermediate tone (5000 Hz) was first delivered as an ambiguous cue, ~25% mice preferred social interaction (positive preference, termed as PP), while ~70% mice chose to step on the platform (negative preference, termed as NP) (sFigure 3g). Again, the activation of pBLA^{Calb1} neurons was more significant in PP group than NP group (sFigure 3h-j) with no change of pBLA^{Calb1} neuron numbers (sFigure 3k). These data further support the involvement of pBLA^{Calb1} neurons in the positive interpretation for ambiguity. We have supplemented these new data in the revised version (please see sFigure 3, page 7 line 18 to line 29).

Figure 3. The pBLA^{Calb1} neurons are robustly activated in positive preference mice exposed to ambiguity. (a) Schematic of a go/go paradigm, in which the positive CS tones (2000 Hz, linked to social reward) and negative CS tones (9000 Hz, linked to foot shocks) were counterbalanced across animals. (b) The mice learnt to associate the positive CS tones with social reward during RC training. $n = 20$ mice per group. Repeated measures two-way ANOVA, [Session 1]: $F(3, 57) = 16.08$, $P < 0.0001$; [Session 2]: $F(3, 57) = 24.01$, $P < 0.0001$; [Session 3]: $F(3, 57) = 20.49$, $P < 0.0001$; [Session 4]: $F(3, 57) = 35.54$, $P < 0.0001$, Tukey's multiple comparisons test, $*P < 0.01$ vs Pre CS (ITI). (c) The mice learnt to associate the negative CS tones with the foot shocks. 10 trials a day (except 10-20 trials on the first day). $n = 20$ mice per group. (d) Mice experienced FC training showed fewer latency and spent more time staying on the platform. (e, f) Mice experienced both RC and FC training showed correct discrimination behaviors in FC test (e) and RC test (f) in response to the CS tones, $n = 20$ mice per group. Paired t test, [step-on%]: $t = 16.46$, $df = 19$, $P < 0.0001$; [poking%]: $t = 26.39$, $df = 19$, $P < 0.0001$. $**P < 0.01$ vs No CS. (g) Proportion of positive

preference (PP), negative preference (NP) and no response (NR) mice experienced the go/go paradigm. **(h-j)** The number of c-Fos (red) and Calb1 (green) co-stained neurons increased in the pBLA of PP group compared with NP group **(h-j)** with similar total Calb1-neuron numbers **(h, k)**. Scale bar, 100 μ m. n = 6 mice per group. Unpaired *t* test, $t = 5.283$, $df = 10$, $P < 0.0004$ **(i)**; $t = 6.641$, $df = 10$, $P < 0.0001$ **(j)**, $t = 0.06591$, $df = 10$, $P = 0.9487$ **(k)**. ** $P < 0.01$ vs NP. Data were presented as mean \pm SEM.

b) in the introduction, it would be important to explain the rationale for investigate the role of the infralimbic cortex. indeed, the introduction provides argumentation to focus on the posterior BLA, but o rationale for the infralimbic cortex is provided in this section.

[Response]: Thanks to the reviewer for the constructive suggestion. We have now added a brief text to overview the rationale for investigating on the role of the infralimbic cortex in the “Introduction” part (please see page 3 line 26 to 29 and page 4 line 1 to 7).

c) do the MPI and LPI subgroups display differential sensitivity to CUMS?

[Response]: Thanks to the reviewer for the question. Indeed, the MPI and LPI subgroups may have differential sensitivity to CUMS due to their different degree in pBLA^{Calb1} neuron activation. The related data supporting this idea are as follows: (1) the pBLA^{Calb1} neurons were less activated in LPI group than MPI group (Figure 1); (2) targeting pBLA^{Calb1} neurons in LPI and MPI significantly reversed their cognitive preference in response to ambiguity (sFigure 5); (3) pBLA^{Calb1} neurons had a monosynaptic innervation from IL neurons (Figure 3); (4) CUMS significantly suppressed IL-pBLA circuit simultaneously with depression and anxiety (Figure 4); and (5) activation of IL-pBLA^{Calb1} circuit dramatically rescued depression- and anxiety-related behaviors in CUMS mice (Figure 5).

To further address the reviewer’s concern, we did additional experiments by conducting a 5-week CUMS paradigm to examine the sensitivity of MPI and LPI to

CUMS. Compared to LPI mice, MPI mice spent more time in the center zone of open field test (sFigure 8a-c) and the open arm of elevated plus maze (sFigure 8d-f). In sucrose preference test, more consumed sucrose than water was detected in MPI mice (sFigure 8g, h). These data indicate that the degree of positive cognition preference correlates well with the resistance to chronic unpredictable stress. We have added these new data in the revised version (please see sFigure 8 and the related descriptions on page 10 line 22 to 27).

Thanks again for the reviewer's insightful comments. Now, we feel that these supplements have greatly strengthened the logic derivation of our research and added to the integrity of our study in terms of susceptibility to chronic unpredictable stress.

sFigure 8. Positive cognition preference correlates better with the resistance to CUMS than the negatives.

(a, d, g) Schematic of open field test (OFT, a), elevated plus maze (EPM, d) and sucrose preference test (SPT, g). MPI-CUMS mice showed less anxiety and depression, evidenced by greater center entries (b) and time staying in the center (c) in OFT, more open-arm exploration (e) and shorter open-arm entry latency (f) in EPM,

and higher sucrose preference in SPT (**h**), while these anti-anxiety and anti-depression effects cannot be detected in LPI-CUMS mice. One-way ANOVA, n=10 mice per group. In OFT: [Center entries] $F(2,27) = 6.920$, $P = 0.0037$; [Time in center] $F(2,27) = 10.23$, $P = 0.0005$. In EPM: [Open arm entries] $F(2,27) = 8.206$, $P = 0.0016$; [open-arm entry latency] $F(2,27) = 13.43$, $P < 0.0001$. In SPT: $F(2,27) = 14.87$, $P < 0.0001$. Bonferroni post hoc analysis, $**P < 0.01$ vs Con, $\# P < 0.05$ vs LPI. Data were presented as mean \pm SEM.

d) in all experiments using optogenetics or chemogenetics, it is mandatory to provide data showing the optogenetic/chemogenetic excitation or silencing actually changed the activity pattern of the targeted area or projection. That is: does excitation of the pBLA really excite pBLA cells and similarly does inhibition of the pBLA inhibit pBLA cells? and so on for all experiments presented. Without this control, authors cannot claim that actually they did excite or inhibit the areas or the projections. This is an important concern. This is necessary to reach the standard of the field.

[Response]: We deeply appreciate the reviewer for pointing out this important issue. As suggested, we had supplemented electrophysiological recordings to confirm the activity changes of the targeted area upon optogenetic/chemogenetic excitation or silencing. We have added these new data to the revised paper (please see Figure 2o, t, and sFigure 4c, h, sFigure 5c, h). These data demonstrate the efficiency of our target manipulation.

e) in the same vein as concern d), in all experiments using viral injections (optogenetics, chemogenetics), figures showing the area of the AAV expression 4 weeks after the injection should be provided, showing its extension in the targeted area, as well as non expression in adjacent areas. This is necessary to reach the standard of the field.

[Response]: Thanks to the reviewer for pointing out this important issue. As suggested, we had supplemented figures of target sites (please see Figure 2n, s; Figure 3i, q, t; sFigure 4a, f; and sFigure 5a, f). These data further confirmed our precise

targeting and greatly supported our conclusions from target manipulation experiments.

f) did data fit normality of distribution and homogeneity of variance? if not, Anovas cannot be used

[Response]: Thanks to the reviewer for the suggestion. We have re-examined the data and found that all the data fit normality of distribution and homogeneity of variance. The method for checking the normality of distribution and homogeneity of variance has been added into the revised method (please see page 23 line 1-4).

g) the CUMS protocol includes food or water deprivation lasting 24 hr. This will interfere with sucrose drinking and with exploratory behaviour. So a table should be provide showing a timeline with the stressors, and the moment of the behaviour testing, in order to check absence of interference.

[Response]: As suggested, we have provided Tables to detail the CUMS paradigm and a timeline with the moment of the behavior testing in the revised version (please see page 21 line 10-12; Figure 5; and sTable 1).

Figure 5. Activating IL-pBLA^{Calb1} circuit ameliorates anxiety and depression behaviors in CUMS mice. (a) Schematic of IL-pBLA photostimulation and pBLA^{Calb1} inhibition in CUMS mice during OFT, EPM and SPT. AAV-CaMKII-ChR2 and AAV-DIO-hM4Di were injected into the IL and pBLA of Calb1-Cre mice, respectively. At the 5th week of CUMS training, fibers were implanted in the pBLA. 30 min before behavior tests, CNO was intraperitoneally injected. ChR2-CUMS mice were tested in a single 9 min session in OFT or EPM with three 3-min epochs. (b, c, e-g, h) ChR2-CUMS mice showed less anxiety and depression behaviors during the illumination epoch, evidenced by greater center entries (b) and time staying in the center (c) in OFT, more open-arm exploration (e, f) and shorter open-arm entry latency (g) in EPM, and higher sucrose preference in SPT (h), while these anti-anxiety and anti-depression effects were reversed when Calb1+ neurons in the

pBLA were inhibited by CNO injection. No difference on travelled distance was detected during and after light stimulation or CNO injection (**d**). Two-way ANOVA group \times epoch interaction, $n=11$ mice per group. In OFT: [Center entries]: $F(4,90) = 3.971, P = 0.0052$; [Time in center]: $F(4,90) = 5.510, P = 0.0005$; [Distance moved]: $F(2,60) = 0.3377, P = 0.7174$. In EPM: [Open arm entries]: $F(4,75) = 2.581, P = 0.0439$; [Time in open arm]: $F(4,81) = 9.084, P < 0.0001$; [open-arm entry latency]: $F(4,75) = 3.344, P = 0.0142$. [In SPT]: $F(2,48) = 34.86, P < 0.0001$. Bonferroni post hoc analysis, $**P < 0.01$ vs eYFP:IL-pBLA+veh:pBLA; $## P < 0.01$ vs Chr2:IL-pBLA+veh:pBLA. Data were presented as mean \pm SEM. (**i**) Proposed working model. The excitatory IL-pBLA^{Calb1} circuit drives interpretation for ambiguous information in a bidirectional and reversible manner. Inhibiting IL-pBLA^{Calb1} pathway produces pessimistic phenotypes, while activating the circuit promotes positive interpretations for the ambiguity, exerting anti-anxiety and anti-depression effects.

sTable1

Week	Day	Food deprivation	Water deprivation	Foreign object exposure	Cold exposure	Illumination	Restraint
1	1	24h (6:00 a.m.-6:00 a.m. on day 2)	24h (6:00 a.m.-6:00 a.m. on day 2)				
	2			1 h (5:00 a.m.-6:00 a.m.)			2 h (6:00 p.m.-8:00 p.m.)
	3				2 h (9:00 a.m.-11:00 a.m.)	12 h (6:00 p.m.-6:00 a.m. on day 4)	
	4				2 h (7:00 a.m.-9:00 a.m.)		2 h (4:00 p.m.-6:00 p.m.)
	5	24 h (6:00 a.m.-6:00 a.m. on day 6)		1 h (7:00 p.m.-8:00 p.m.)			
	6		24h (6:00 a.m.-6:00 a.m. on day 7)			12 h (7:00 a.m.-7:00 p.m.)	
	7			1 h (9:00 a.m.-10:00 a.m.)	2 h (4:00 p.m.-6:00 p.m.)		

Week	Day	Food deprivation	Water deprivation	Foreign object exposure	Cold exposure	Illumination	Restraint
2	1	24h (6:00 a.m.-6:00 a.m. on day 2)					2 h (6:00 p.m.-8:00 p.m.)
	2		24h (6:00 a.m.-6:00 a.m. on day 3)			12 h (6:00 p.m.-6:00 a.m. on day 3)	
	3			1 h (5:00 a.m.-6:00 a.m.)	2 h (9:00 a.m.-11:00 a.m.)		2 h (4:00 p.m.-6:00 p.m.)
	4	24 h (6:00 a.m.-6:00 a.m. on day 5)		1 h (7:00 p.m.-8:00 p.m.)			
	5				2 h (7:00 a.m.-9:00 a.m.)		2 h (4:00 p.m.-6:00 p.m.)
	6		24h (6:00 a.m.-6:00 a.m. on day 7)	1 h (4:00 p.m.-5:00 p.m.)			2 h (8:00 p.m.-10:00 p.m.)
	7				2 h (4:00 p.m.-6:00 p.m.)	12 h (6:00 a.m.-6:00 p.m.)	2 h (8:00 a.m.-10:00 a.m.)

Week	Day	Food deprivation	Water deprivation	Foreign object exposure	Cold exposure	Illumination	Restraint
3	1		24h (6:00 a.m.-6:00 a.m. on day 2)				2 h (6:00 p.m.-8:00 p.m.)
	2	24h (6:00 a.m.-6:00 a.m. on day 3)		1 h (5:00 a.m.-6:00 a.m.)	2 h (4:00 p.m.-6:00 p.m.)		
	3					12 h (6:00 p.m.-6:00 a.m. on day 4)	2 h (4:00 p.m.-6:00 p.m.)
	4		24h (6:00 a.m.-6:00 a.m. on day 5)				2 h (4:00 p.m.-6:00 p.m.)
	5			1 h (7:00 a.m.-8:00 a.m.)	2 h (4:00 p.m.-6:00 p.m.)	12 h (6:00 a.m.-6:00 p.m.)	
	6	24h (6:00 a.m.-6:00 a.m. on day 7)			2 h (9:00 a.m.-11:00 a.m.)		2 h (3:00 p.m.-5:00 p.m.)
	7			1 h (1:00 p.m.-2:00 p.m.)	2 h (4:00 p.m.-6:00 p.m.)		2 h (8:00 a.m.-10:00 a.m.)

Supplemental Table 1: Experimental Stressors and Scheduling for Chronic

Unpredictable Mild Stress (CUMS). Mice were exposed to a variety of mild unexpected stressors for 6 weeks, including water and/or food deprivation, foreign object exposure, restraint, cold exposure, and reversal of the light/dark cycle. The procedures were repeated from week 1 after first 3 weeks. Some stressors were interrupted or modified when they occurred at particular points in the sucrose-preference test schedule.

Reviewer #2 (Remarks to the Author):

Yu and colleagues have used a series of behavioral, recording, and projection-specific manipulation studies to examine the role of IL->BLA calbindin cells in modulating behavioral responses related to reward and aversion. While the use of several advanced tools in combination is impressive, the overall rationale for the studies and interpretations presented are problematic for several reasons.

[Response]: We deeply appreciate the reviewer for the positive comments and constructive suggestions to our paper.

The premise of the model used, as well as interpretations throughout, are overly anthropomorphic. The paper is framed around an idea that we can understand cognitive attributions of mice as being optimistic or pessimistic, and further, it attempts to link anxiety and depression to the findings presented. This is entirely inappropriate for studies using rodents. The language of the paper and framing of the data need to be completely overhauled.

[Response]: We thank the reviewer for bringing up this issue. Indeed, due to the inability of humans and rodents to communicate through language, the self-reporting test from human/patients that is critical for diagnosis of optimistic and/or pessimistic bias cannot be completed by mice. Nonetheless, a number of mouse behavioral paradigms have been developed and widely applied to the field of cognition and emotion, such as dementia¹²⁻¹⁴, attention¹⁵, anxiety^{16, 17}, depression^{7, 18}, aggression¹⁹, and decision⁹, etc. Our design for the go/go behavioral paradigm of cognitive preferences was inspired by Harding's work²⁰. In Harding's go/no-go task of ambiguous-cue interpretation, rats were first trained to press a lever in the presence of a tone associated with a positive event (delivery of a food pellet) and to refrain from pressing the lever as a way to avoid a negative event (white noise) associated with another tone. Then, an ambiguous tone was delivered to probe animals' relative anticipation of positive and negative events. Behavior responses such as few and slow lever press were used as indicators of negative anticipation in response to ambiguity.

For unpredictably-stressed rats, these behavioral responses were obviously detectable, indicating the existence of “pessimistic” response bias²⁰.

In the present study, we develop a go/go task of ambiguous-cue interpretation by introducing CS-associated nose poking behavior and avoidance behavior. We modified the go/no-go task for two reasons: (1) the go/no-go task paradigm may not distinguish whether a response bias originates from a reduced positive and/or an increased negative responding; and (2) a ‘no-go’ as a response indicator cannot be distinguished from a response omission. Using our novel go/go task, we identified PP mice via their poking behaviors and NP mice through their active stepping-on response in the presence of ambiguous tone.

We hope the reviewer could agree that our novel go/go paradigm on mice is meaningful for understanding the phenomenon of cognitive preference in response to ambiguity, though further exploration or modification for better paradigms to replicate human optimistic and pessimistic bias on the experiment animals is needed in future studies.

Additionally, we have detailed information and reframed the data by providing several additional Figures and Tables, and have asked professional editing services for proofreading of the manuscript (SpringerNature Author Services: 95E4-BB00-7AC1-079C-57AA). We feel that the quality of manuscript has been substantially improved.

Second, what's described as 'optimistic or pessimistic preference' seems to actually be basic generalization and seemingly random/guessing behavior to an ambiguous cue. There have been many studies in the field on cue generalization and ambiguity for both appetitive and aversive learning, as well as studies combining appetitive and aversive cues in the same task. None of these are cited or mentioned. Rather, the simple phenomenon of generalization is framed as an inherent mouse cognitive bias - a much less parsimonious explanation for the observed behavior. Before the data here can be properly interpreted, I suggest the authors compare their findings to these related studies in the field and present them using the language already established in

the field.

[Response]: We thank the reviewer for the suggestions. As suggested, we have discussed that matter and cited the relative references in the revised paper (please also see page 14 line 26-28).

Due to the heritability^{21,22}, both optimism and pessimism may probably be viewed as inherent traits of individuals. However, there are other evidence revealing that these cognitive biases can be modified and shift from each other when the individual encounters controllable and/or uncontrollable stressful situations²³. Therefore, with the richness of experience, individual's trait could be dynamically shaped, consequently forming optimistic or pessimistic preference. Combined with the existence of cognitive bias in rodents²⁰, we hope the reviewer might agree that optimism and pessimism should not be recognized as fixed and unchangeable traits, on the contrary, these cognition biases and their alterations deserve careful study and can be investigated in rodents especially experienced different stresses.

Since natural stimuli rarely occur in the exact same form from one encounter to the next, the ability to generalize learning across stimuli and across situations is essential. For example, generalization of fear learning is similarly adaptive, as it enables avoidance of threats not specifically encountered before. However, excessive generalization is maladaptive if it leads to fear responses that are too strong or that occur in inappropriate situations, interfering with more rewarding actions. Inappropriate fear is the cardinal feature of many mental health disorders, such as anxiety disorders, obsessive compulsive disorder, and PTSD^{24,25}. Thus, generalization should not be a byproduct of conditioned learning, on the contrary, it can be an active process in which expressed behaviors are determined by cognitive preference for the ambiguity based on previous experiences. Given that, we developed go/go paradigm to identify generalized mice with either high negative or positive cognitive preference for ambiguous information. Then, we dissected a novel IL-pBLA^{Calb1} circuit and demonstrated its contribution to positive interpretation for the ambiguity. Further, targeting IL-pBLA^{Calb1} circuit was found to significantly ameliorate anxiety and depression in CUMS model.

Other minor issues:

- In the RC task, mice are defined as more/less positive interpretation (MPI or LPI) depending on whether they respond to the ambiguous cue. MPI mice are simply mice that generalize more than LPI mice.

[Response]: Yes, we agree with the reviewer that the MPI mice are more generalized than the LPI mice. We used the terms of “more positive interpretation” and “less positive interpretation” instead of “high generalization” and “less generalization” based on two reasons: (1) the contribution of more positive cognition to higher reward generalization, and (2) the main theme of the present study, i.e., improvement of positive cognition on anxiety and depression after chronic unpredictable stress.

- With regard to negative preference (NP) and positive preference (PP) mice, with a ratio of NP:PP close to 50:50, this seems to further imply that what's going on here is random sampling/choice.

[Response]: We apologize for the misleading information caused by the insufficient description. In the present study, NP and PP mice were clearly identified by go/go paradigm for ambiguous information based on their first response to the first ambiguous tone. We found that this first choice was positively correlated with the subsequent 9 choices during the ambiguous tone probe (sFigure 2). In another word, if the mice chose to poke for the reward at the first time, the probability of poking was much greater than that of stepping-on behavior for shock avoidance in the next 9 trials during ambiguous tone probe (sFigure 2a), and vice versa (sFigure 2b). Therefore, a ratio of NP:PP close to 50:50 was not random sampling/choice, but, represented ratio of mice with different cognitive preference for the ambiguous information. We have emphasized this point with supplemented data in the revised version (please see sFigure 2, page 7 line 11-16).

Figure 2. The first choice is positively correlated with the subsequent 9 choices during the ambiguous tone probing. For both positive preference (PP, **a**) and negative preference (NP, **b**) mice, the choice proportion was calculated in the next 9 trials in response to the ambiguous tone.

Reviewer #3 (Remarks to the Author):

In this manuscript, the authors investigate the role of the posterior basolateral amygdala Calbindin 1 expressing neurons (pBLACalb1), and the infralimbic (IL) - pBLACalb1 circuit in positive interpretation bias of an ambiguous cue in a go/go task that the group developed. The authors first use cFos staining to identify increased activity in IL and pBLACalb1 neurons following a short-latency nose poke for reward to a novel cue after initial reward conditioning to a different cue. The authors then show that cFos increases in the pBLACalb1 in a go/go paradigm, after a positive response (nose poke as for a reward cue rather than platform avoidance as for an aversive cue) to a novel cue, after training on both reward and fear conditioning with two different cues. The authors then show that optogenetic excitation and inhibition of IL input to pBLA bidirectionally increases and decreases, respectively, the positive response rate and latency (nose poke) to a novel cue in the go/go task. Finally, the authors show that after chronic unpredictable mild stress, optogenetic activation of the IL input to the pBLA ameliorates anxiety- and depression-like behaviors in a BLACalb1 – dependent manner. The authors conclude that anxiety and depression like symptoms can be ameliorated by activation of the IL-pBLA Calb1 circuit, thereby suggesting that this pathway is vital for biasing an animal towards optimistic behaviors.

Overall, this manuscript presents interesting data that develop ideas regarding conflicting cue interpretation. Furthermore, the authors develop an interesting go/go paradigm to investigate these questions. However, the manuscript suffers from substantial organizational and grammatical problems that interfere with its ability to fully develop the data. In addition, I have some concerns, described below, regarding the controls used in this study, and the claims that the authors make regarding pessimism and optimism (although this is perhaps a reflection of the language difficulties described above).

[Response]: We deeply appreciate the reviewer for the positive comments and constructive suggestions.

Major

- Related to the issue of language, there is misuse of tense, plurality and irregular sentence structure that appear throughout the paper (e.g. paragraph 1 of the introduction, etc). In addition there is quite a bit of anthropomorphizing in the text (e.g. “BLA neurons with their associated circuits interpret...cues”, etc.).

[Response]: We sincerely apologize for the poor writing. As suggested, we have asked professional editing services for proofreading of the manuscript (SpringerNature Author Services: 95E4-BB00-7AC1-079C-57AA). We feel that the flow and language level have been substantially improved.

- The behavioral data are not sufficiently described to give an understanding of what is being studied (e.g. optimism or pessimism, or positive preference vs negative preference).

[Response]: We are very sorry for the missing information. We have added these details in the revised version (please see the followings and red part in the revised revision).

For example, the reward and aversive nature of the cues aren't established as equally salient for the animals. The 30-s reward cue was paired with sucrose over 16 days of training (with 4 different ITI trained for 4 days each), whereby nose poke to the cue and consumption of reward terminated the trial. (As a side note, the authors show data from a recall test that is described as occurring 24 hrs after training, but it occurs after the animals have been learning and recalling this training for weeks).

[Response]: We apologize for the misleading. Actually, our recall test was performed at 17th day, i.e., 24 h after the last training trial of session 4, not 24 h after the first training trial. We carefully checked the manuscript and corrected ambiguous expressions in the results (please see page 6 line 8).

However, from the description given in the methods, it seems that the fear

conditioning employed a different approach, whereby animals were shocked for 10 sec at a time during the duration of the aversive cue (0.25mA), unless they stepped on the avoidance platform. The only behavioral analysis of the fear conditioning paradigm shown is in fig 1c, where Times of Step-On across 7 days of training is shown. Is this the average number of times that the animal stepped onto the platform during trials on a training day?

[Response]: We are very sorry for the missing information. In Figure 2c, it is the times (presented as mean \pm SEM) of correct step-on responses upon CS during FC training. In our present study, FC training lasted for 7 days, 10 trials per day (except 10-20 trials on the first day). At the beginning, only few step-on behavior was associated with CS (Figure 2c and sFigure 1). As FC training went on, CS-associated step-on response was significantly increased, indicating the improvement in FC learning (Figure 2c and sFigure 1). At 6th and 7th day of FC training, the correct response was 10 in response to 10 CS, which means 100% correct rate (Figure 2c). Meanwhile, the staying time on the platform was more than 30 s, and the step-on latency was about or less than 2s (sFigure 1). In the revised paper, we detailed FC training in the part of methods (please see page 20 lines 17-20) and the supplemented new data to accurately present progressive learning ability during FC training (please see sFigure 1).

sFigure 1. Mice gradually learn to associate the negative CS tones with foot

shocks. The step-on latency was gradually decreased and the staying time on the platform increased along with FC training. Red and black represent the mice with and without stepping-on response (SOR), respectively. 10 trials a day (except 10-20 trials on the first day). n = 20 mice per group.

According to the methods, the fear conditioning sessions would be different lengths on different days, because they would stop after animals stayed on the platform for at least 30 seconds.

[Response]: We understand the reviewer's concern, and we fully agree with the reviewer that ITI in FC training should be set different. We actually had done the experiments that way. However, due to the poor English expression, we failed to articulate the details of the FC training in the previous version of the paper.

To avoid poking behavior by chance, we increased ITI gradually along the four RC training sessions. When ITI exceeded CS duration (session 4), more poking behavior during CS than ITI, as shown in Figure 1b and Figure 2b, strongly indicated the success of RC training.

For FC training, each trial consisted of 10 s-CS (except 10-20 trials on the first day) and at least 30 s-staying time on the platform. Thus, there was impossible for the mice to step on the platform by chance. Considering the variation of step-down latency of each mouse during each training trial, we changed ITI randomly from 45 s to 60 s among 10 trials of each day. As shown in Figure 2c, a good correlation of CS with aversive shock was successfully established after 7-day FC training. We have supplemented these details in the revised version (please see page 20, line 17-20).

We totally agree with the reviewer that equally salient of FC and RC for the animals may benefit a lot for the investigation on cognition preference in the presence of ambiguity. In addition to training days, the nature and strength of stressors during FC/RC training may also contribute to behavior responses in face of ambiguous cue. As shown in Figure 2f, the ratio of positive and negative preferences was ~1 to 1 in response to the ambiguity, but when sucrose reward was placed by social reward (sFigure 3a), the ratio changed into ~1 to 3 (sFigure 3g). Thus, we hope the reviewer

will agree that the present FC training is acceptable and appropriate for go-go paradigm.

Figure 3. The pBLA^{Calb1} neurons are robustly activated in positive preference mice exposed to ambiguity. (a) Schematic of a go/go paradigm, in which the positive CS tones (2000 Hz, linked to social reward) and negative CS tones (9000 Hz, linked to foot shocks) were counterbalanced across animals. (b) The mice learnt to associate the positive CS tones with social reward during RC training. $n = 20$ mice per group. Repeated measures two-way ANOVA, [Session 1]: $F(3, 57) = 16.08$, $P < 0.0001$; [Session 2]: $F(3, 57) = 24.01$, $P < 0.0001$; [Session 3]: $F(3, 57) = 20.49$, $P < 0.0001$; [Session 4]: $F(3, 57) = 35.54$, $P < 0.0001$, Tukey's multiple comparisons test, $*P < 0.01$ vs Pre CS (ITI). (c) The mice learnt to associate the negative CS tones with the foot shocks. 10 trials a day (except 10-20 trials on the first day). $n = 20$ mice per group. (d) Mice experienced FC training showed fewer latency and spent more time staying on the platform. (e, f) Mice experienced both RC and FC training showed correct

discrimination behaviors in FC test (**e**) and RC test (**f**) in response to the CS tones, $n = 20$ mice per group. Paired t test, [step-on%]: $t = 16.46$, $df = 19$, $P < 0.0001$; [poking%]: $t = 26.39$, $df = 19$, $P < 0.0001$. ** $P < 0.01$ vs No CS. (**g**) Proportion of positive preference (PP), negative preference (NP) and no response (NR) mice experienced the go/go paradigm. (**h-j**) The number of c-Fos (red) and Calb1 (green) co-stained neurons increased in the pBLA of PP group compared with NP group (**h-j**) with similar total Calb1-neuron numbers (**h, k**). Scale bar, 100 μm . $n = 6$ mice per group. Unpaired t test, $t = 5.283$, $df = 10$, $P < 0.0004$ (**i**); $t = 6.641$, $df = 10$, $P < 0.0001$ (**j**), $t = 0.06591$, $df = 10$, $P = 0.9487$ (**k**). ** $P < 0.01$ vs NP. Data were presented as mean \pm SEM.

Was the sucrose also available at this time as an incentive to step off the platform to go back to the nose poke? If so, did the mice use the ITI to press for sucrose?

[Response]: Thanks to the reviewer for the questions. During FC/RC test, poking or avoiding was an exclusive choice in each specific CS trial. For example, as soon as the animal chose to step on the platform in response to the negative CS tone, no sucrose would be available during the whole trial, because that the infrared beam-coupled sucrose delivery was positive CS dependent. A more complicated case was that the animal chose to step on the platform in response to the positive CS tone. If it happened, we switched off the sucrose delivery system manually. Therefore, our go/go paradigm was different from classic platform avoidance training^{26, 27} and it's impossible for the mice to use the ITI to poke for the sucrose reward. We have added these details in the revised version (please see page 20 line 28 to 31).

Were the animals also showing defensive freezing? How long of each trial did the animals spend on the platform? It would be helpful to analyze these data to reflect training conditions (e.g. number of trials per day, latency to step on the platform), and to help understand the behavior in the go-go paradigm.

[Response]: Thanks to the reviewer for the constructive suggestions. Each FC training day consisted 10 trials (except 10-20 trials on the first day). When negative CS was on,

mice were trained to step on the platform to avoid foot shock. Since our step-on platform (3.0 cm high and 5.0 cm wide) was relative smaller than the traditional size²⁸, here was no extra spaces for mouse to move freely on the platform. Therefore, it is hard to accurately identify defensive freezing during FC training. Instead, we recorded total staying time on the platform (sFigure 1). After 7-day FC training, most mice could stay on the platform for ~40 s. The latency to step on the platform was dramatically decreased and less than ~2 s at 7th day of FC training (sFigure 1). To help better understanding of the FC behaviors in the go-go paradigm, we have detailed the description and added a new figure in the revised version (please see sFigure 1 page 20 line 8-22).

sFigure 1. Mice gradually learn to associate the negative CS tones with foot shocks. The step-on latency was gradually decreased and the staying time on the platform increased along with FC training. Red and black represent the mice with and without stepping-on response (SOR), respectively. 10 trials a day (except 10-20 trials on the first day). n = 20 mice per group.

There is a body of work on fear conditioning and platform avoidance training from Greg Quirk's lab, that this manuscript doesn't make contact with, but which would be helpful for interpretation.

[Response]: We appreciate the reviewer's suggestion, and we have carefully reviewed works from Greg Quirk's lab and cited the papers in our revised version (please see page 4 line 19 and page 20 line 29).

- In several instances it is not clear what controls are being used. For example, the reward conditioning experiment is described in the text as having a no-CS group (line 89). This suggests that there are 2 groups of animals, one that receive CS pairing with reward, and one that was in the box but didn't receive a CS. However, figure 1b shows behavior during the CS+ (black squares) and Pre-CS behavior (grey squares), suggesting that there is 1 group of animals and behavior was scored during the CS, and pre-CS behavior was used as the control condition. Figure 1c refers to a CS- (grey) and a CS+ (black), suggesting that there was a CS that wasn't paired with anything and a CS that was paired with reward. The description in the methods section does not provide any further information.

[Response]: We are very sorry for not being able to clearly presented. Actually, we used one group of mice for RC training (Figure 1b) and testing (Figure 1c-f). The pre-CS behavior during RC training was used as the control condition. After training, the same cohort of mice was tested in the training box and the novel box, in which no-CS behavior was used as the control of CS condition (Figure 1c). After that, these mice were transferred back to training box, and their behavioral responses to the CS tone and novel tone were monitored and compared between these two different tones (Figure 1d-f). In the revised version, we have clarified these matters by correcting the inappropriate labeling in Figures (please see Figure 1c, Figure 2d, e, and sFigure 3).

- For the fiber photometry experiment shown in figure 3, the authors only analyze data from positive preferring mice without controlling for data for negative preferring mice. Did the negative preferring mice show similar changes, or was the IL-pBLA circuit less active in those animals?

[Response]: Thanks to the reviewer for pointing out this issue. We performed in vivo calcium recording on NP mice as what we did on PP group. Compared with baseline, an increased calcium activity of IL-pBLA circuit was detectable when an intermediate tone was on (Figure 3k, m). Compared with NP group, the average peak calcium activity before poke was much higher in PP group (Figure 3n), and the increased slope for calcium transit in PP group was much greater than the NP group (Figure 3o). No statistically significant fluctuation in calcium activity was detected in IL-pBLA-GFP control group (Figure 3n and sFigure 6). These data suggest that the activation of IL-pBLA circuit encodes positive preference for the ambiguity. Also, more rapid and robust activation of IL-pBLA circuit are indicated to be correlate with stronger positive cognition preference for the ambiguity. We have clarified these in the figure and the text of the revised paper (please see Figure 3k, m-o; page 9 line 20-26).

Figure 3. IL-pBLA inputs trigger positive interpretation for the ambiguity.

(a, b) Schematic and representative image of CTB retrograde tracing (scale bar=100 μm). **(c, d)** Schematic and representative image of HSV anterograde tracing (scale bar=50 μm). **(e)** Representative images of patch pipette tips on pBLA^{Calb1+} (tdTomato+) neurons in brain slices (scale bar=30 μm). **(f)** Representative traces of EPSCs in the IL–pBLA^{Calb1+} pathway recorded under different experimental conditions. EPSCs were evoked by photostimulating ChR2-expressing axons from IL, and recorded in Calb1⁺ neurons of the pBLA. Optogenetically-induced and tetrodotoxin (TTX)-blocked EPSCs were partially rescued by 4-aminopyridine (4-AP), indicating monosynaptic nature of the connections in the IL to pBLA^{Calb1+} pathway. **(g)** Changes of EPSCs amplitude in TTX only and TTX + 4-AP (n=13 pBLA^{Calb1+} neurons from 6 IL–pBLA–ChR2 mice). **(h)** Schematic of Ca²⁺ recording of IL–pBLA circuit in the go/go paradigm. **(i)** Representative confocal image of GCaMP6 expression in IL neurons projecting to the pBLA (IL–pBLA neurons). Scale bar, 100 μm . **(j, k)** Heat maps of normalized Ca²⁺ activity in IL–pBLA neurons when the intermediate tones were present in PP group **(j)** and NP group **(k)**. **(l, m)** Average Ca²⁺ transients from IL–pBLA neurons in response to intermediate tones in PP group and NP group (n =8 mice per group). **(n)** Average df/f for baseline and interpretation period in response to the intermediate tones in PP/NP-GCaMP6f mice and PP/NP-eGFP mice. Two-way ANOVA, n = 8 or 9 mice per group. $F(3,61) = 142.4$, $P < 0.0001$, Bonferroni post hoc analysis, $**P < 0.01$ vs Baseline, $## P < 0.01$ vs NP. **(o)** Compared with NP group, the rise slope for calcium transit increase was greater in PP group. Unpaired t test, $t = 4.702$, $df = 10$, $P = 0.0008$, n= 8 mice per group. $**P < 0.01$ vs NP. **(p)** Schematic showing light illuminating epoch when the intermediate tone was delivered during go/go paradigm. **(q, t)** The representative confocal images of NpHR and ChR2 expression in the IL–pBLA neurons after stereotaxic infusion. Scale bar=100 μm . **(r, s)** Inhibiting pBLA^{Calb1} decreased the proportion of PP and prolonged the first poking latency of NpHR-PP mice. **(u, v)** Activating pBLA^{Calb1} decreased the proportion of NP and shortened the first poking latency of ChR2-NP mice. Two-way ANOVA group \times epoch interaction; first poking latency [NpHR-PP] $F(1,34) = 6.969$, $P = 0.0282$, Bonferroni post hoc analysis, $*P < 0.05$ vs eYFP: IL–pBLA light on;

[Chr2-NP] $F(1,46) = 6.085$, $P = 0.0163$, Bonferroni post hoc analysis, $*P < 0.05$ vs eYFP: IL-pBLA light on. $n = 9\sim 15$ mice per group. Data were presented as mean \pm SEM.

-similar to the point above, in experiments shown in figure 2, the IL input to the pBLA was inhibited only in the positive preferring mice, and stimulated only in the negative preferring mice. Did inhibition in negative preferring mice and stimulation in positive preferring mice not affect behavior?

[Response]: Thanks to the reviewer for raising this question. Compared with NP group, IL-pBLA^{Calb1} pathway was significantly activated in PP group (please see Figure 2 and 3). Therefore, we inhibited the pathway in NP mice and activated the pathway in PP mice in the following studies. By these targeting, the reversal phenotypes of cognition interpretation in NP and PP groups demonstrated an essential role of IL-pBLA^{Calb1} pathway in the positive cognition preference.

Given the difference in the activation degree of IL-pBLA^{Calb1} pathway between NP and PP (Figure 3), we speculate that further activation in PP group or inhibition in NP group may induce “ceiling effect” or “floor effect” of IL-pBLA^{Calb1} pathway to the positive cognition preference in response to the ambiguity. We believe that this interesting question deserves an independent investigation in the future.

- In several instances the authors use unpaired t-test statistics where rm-ANOVAs, one-way ANOVAs, or two-way ANOVAs would be more appropriate (e.g. fig 1e, f, 1h, etc).

[Response]: Thanks to the reviewer for the suggestion. We have carefully checked statistics and used paired t-test statistics for Figure 1c, e, f, Figure 2 d, e, sFigure 3e, f, and rm-ANOVAs for Figure 1b, Figure 2b, sFigure 3b.

- In figure 2k, the authors show that in positive preferring (nose-poking) mice in the go/go task, the Calb1+ cells make up a larger percentage from the total active pool (Calb1:cfos/cfos) over the negative preferring mice. Does the percent of active Calb1

cells also increase (Calb1:cfos/Calb1)?

[Response]: As suggested, we quantified Calb1:cfos/Calb1 and found an increase in PP group compared to NP group (Figure 2I). Inspired by the reviewer, we also added relevant analysis on LPI and MPI (Figure 1I). These data further demonstrated that the positive interpretation for ambiguous cue was associated with robust activation of pBLA^{Calb1} neurons. We have added these new analyses in the revised version (please see Figure 1I and Figure 2I).

Figure 1. Positive interpretation for ambiguous cues is associated with robust activation of pBLA^{Calb1} neurons.

(a) Schematic of reward conditioning (RC) training and test. The mice were trained to associate a CS tone (2000 Hz, lasting 30 s) with sucrose reward. Then, test consisting of 10 trials was conducted in the training box or a novel box. In each test trial, a CS tone and sucrose reward was stopped immediately after mice poked. (b) RC training

induced an increasingly improved association of CS tone with the predicted sucrose. The training was composed of 16 days with ~35 trials per day. The inter-trial intervals (ITIs) were changed every 4 days to form 4 different training sessions. The percentage of poking times per day were calculated. $n = 20$ mice per group. Repeated measures two-way ANOVA, [Session 1]: $F(3, 57) = 56.40$, $P < 0.0001$; [Session 2]: $F(3, 57) = 31.82$, $P < 0.0001$; [Session 3]: $F(3, 57) = 30.49$, $P < 0.0001$; [Session 4]: $F(3, 57) = 33.63$, $P < 0.0001$, Tukey's multiple comparisons test, $**P < 0.01$ vs Pre CS (ITI). **(c)** The CS tones induced an increased poking time in both training box and novel box. $n = 20$ mice per group. Paired t test, [Training box]: $t = 18.7$, $df = 19$, $P < 0.0001$; [Novel box]: $t = 15.38$, $df = 19$, $P < 0.0001$. $**P < 0.01$ vs No CS. **(d)** Schematic for ambiguous cue test. **(e, f)** The CS-trained mice showed fewer poking times and longer first poking latency in response to a novel tone (5000 Hz, ambiguity) than to the CS tone (2000 Hz). $n = 20$ mice per group. Paired t test, [Poking times]: $t = 4.876$, $df = 19$, $P < 0.001$; [First poking latency]: $t = 3.370$, $df = 19$, $P = 0.0032$. $**P < 0.01$ vs CS tone. **(g)** The mice were further divided into more positive interpretation (MPI) and less positive interpretation (LPI) groups according to their different poking times (cutoff: 5 poking times) to the ambiguous tone (**left**), and both MPI and LPI groups had same poking times to the CS tones (**right**). $n = 11$ or 9 mice per group. Unpaired t test, [Novel tones]: $t = 6.959$, $df = 18$, $P < 0.0001$; [CS tones]: $t = 0.3927$, $df = 18$, $P = 0.6996$. $**P < 0.01$ vs MPI. **(h, i)** The MPI and LPI mice showed distinct activation patterns in the pBLA and the other indicated brain regions measured by c-Fos staining. Scale bar, $100 \mu\text{m}$. $n = 4\sim 5$ mice per group. Unpaired t test, [IL]: $t = 3.056$, $df = 8$, $P = 0.0157$; [pBLA]: $t = 2.683$, $df = 8$, $P = 0.0278$; [BMP]: $t = 2.672$, $df = 6$, $P = 0.0369$. $*P < 0.05$ vs LPI. **(j-l)** Activation of pBLA^{Calb1+} neurons was more significant in MPI than LPI group measured by c-Fos (red) and Calb1 (green) co-staining. Scale bar, $100 \mu\text{m}$. $n = 5$ to 7 mice per group. Unpaired t test, $t = 2.440$, $df = 10$, $P = 0.0348$ (k); Unpaired t test, $t = 2.999$, $df = 10$, $P = 0.0134$ (l). $*P < 0.05$ vs LPI. Data were presented as mean \pm SEM.

Figure 2. $pBLA^{Calb1}$ neurons govern positive cognition bias for ambiguous cue.

(a) Schematic of a go/go paradigm, in which the positive CS tones (2000 Hz, linked to sucrose) and negative CS tones (9000 Hz, linked to foot shocks) were counterbalanced across animals. (b) Mice learnt to associate the positive CS tones with sucrose during RC training. $n = 20$ mice per group. Repeated measures two-way ANOVA, [Session 1]: $F(3, 57) = 88.46, P < 0.0001$; [Session 2]: $F(3, 57) = 33.64,$

$P < 0.0001$; [Session 3]: $F(3, 57) = 7.35$, $P < 0.0001$; [Session 4]: $F(3, 57) = 14.72$, $P = 0.0005$, Tukey's multiple comparisons test, $**P < 0.01$ vs Pre CS (ITI). (c) Mice learnt to associate the negative CS tones with foot shocks. $n = 20$ mice per group. (d, e) The mice experienced both RC and FC training showed correct discrimination behaviors in FC test (d) and RC test (e) in response to the CS tones, $n = 20$ mice per group. Paired t test, [Step-on%]: $t = 19.09$, $df = 19$, $P < 0.0001$; [Poking%]: $t = 16.49$, $df = 19$, $P < 0.0001$. $**P < 0.01$ vs No CS. (f) Proportion of positive preference (PP), negative preference (NP) and no response (NR) mice experienced the go/go paradigm. (g, h) Number of total c-Fos-positive neurons increased in the pBLA and IL subsets of PP group compared with NP group. Scale bar, $200 \mu\text{m}$. $n = 3\sim 5$ mice per group. Unpaired t test, [IL]: $t = 3.153$, $df = 8$, $P = 0.0135$; [PL]: $t = 2.773$, $df = 8$, $P = 0.0242$; [aBLA]: $t = 2.636$, $df = 6$, $P = 0.0388$; [pBLA]: $t = 2.916$, $df = 6$, $P = 0.0268$; [PH]: $t = 3.285$, $df = 6$, $P = 0.0167$. $*P < 0.01$ vs NP. (i-l) Number of c-Fos (red) and Calb1 (green) co-stained neurons increased in the pBLA of PP group compared with NP group (i, k, l) with similar total Calb1-neuron numbers (j). Scale bar, $100 \mu\text{m}$. $n = 6$ mice per group. Unpaired t test, $t = 0.6014$, $df = 10$, $P = 0.5610$ (j); $t = 2.294$, $df = 10$, $P = 0.0447$ (k); $t = 4.220$, $df = 10$, $P = 0.0018$ (l). $*P < 0.01$ vs NP. (m, r) Schematic of optogenetic stimulations on NpHR-PP and Chr2-NP mice while intermediate (ambiguous) tones were present. (n, s) Representative confocal images of NpHR (n) and Chr2 (s) expression in the pBLA^{Calb1}. Scale bar, $100 \mu\text{m}$. (o, t) Yellow light illumination blocked evoked spiking of NpHR cells (o), while brief blue light pulses at 20 Hz precisely activated Chr2 cells in the pBLA (t). (p, q) Inhibiting pBLA^{Calb1} decreased the proportion of PP (p) and prolonged the first poking latency in NpHR-PP mice (q). $n = 9$ or 10 mice per group. Two-way ANOVA, $F(1, 34) = 4.744$, $P = 0.0107$, Tukey's multiple comparisons test, $*P < 0.05$ vs pBLA: eYFP light on. (u, v) Activating pBLA^{Calb1} decreased the proportion of NP (u) and shortened the first poking latency in Chr2-NP mice (v). $n = 10$ or 17 mice per group. Two-way ANOVA, $F(1, 50) = 6.897$, $P = 0.0017$, Tukey's multiple comparisons test, $*P < 0.01$ vs pBLA: eYFP light on. Data were presented as mean \pm SEM.

-The introduction presents pieces of information (such as the use of a go/go paradigm) without providing prior justification or explanation as to what it is. Although these measures are explained within the methods section, it would benefit the reader to have a lead in as to why they are the appropriate tasks prior to them being introduced as part of the study design.

[Response]: We deeply appreciate the reviewer for the suggestion. In the revised version, we first reviewed Pavlovian reward/aversion association paradigms, and then introduced our go-go paradigm and its advantages in the study of cognitive preference (please see page 4 lines 9 to 25).

--The introduction makes numerous claims (e.g. regarding the benefit of optimism and the risks of pessimism), but employs citations sporadically to support these claims (e.g. line 58, “Targeting these BLA neurons remarkably supports positive reinforcement”). Without citations, it is unclear which of these ideas are generated from previous research and which are being concluded by the authors.

[Response]: We apologize for the missing citations. As suggested, we have added related reference (please see page 3 line 19).

Other:

-Placement images throughout the manuscript don't show anatomical landmarks to help identify the outlined regions, making it difficult to fully understand location. For example, figure 1h, figures 3b,d,i, m,n; Figure 4i, Supp 1, etc.

[Response]: We agree with the reviewer that histological evidence of virus injection sites is important to demonstrate the targeting manipulation. In the revised version of the manuscript, we provided global view of the targeting area, and the non-expression in adjacent areas. From these figures (Figure 2, Figure 3, sFigure 4 and 5), we could see that the expression of eYFP was respectively confined to the pBLA and IL subregions after the injection, which provides evidence for the precise targeting on pBLA and IL.

- Figure 1h shows examples of cFos expression in the pBLA that was quantified in 1i. However, I don't see any cFos+ cells in figure 1h. Please show a more zoomed in example of the cells.

[Response]: As suggested, we provided a new representative image with high quality in Figure 1h.

Figure 1. Positive interpretation for ambiguous cues is associated with robust activation of pBLA^{Cab1} neurons.

(a) Schematic of reward conditioning (RC) training and test. The mice were trained to associate a CS tone (2000 Hz, lasting 30 s) with sucrose reward. Then, test consisting of 10 trials was conducted in the training box or a novel box. In each test trial, a CS tone and sucrose reward was stopped immediately after mice poked. (b) RC training induced an increasingly improved association of CS tone with the predicted sucrose. The training was composed of 16 days with ~35 trials per day. The inter-trial intervals

(ITIs) were changed every 4 days to form 4 different training sessions. The percentage of poking times per day were calculated. $n = 20$ mice per group. Repeated measures two-way ANOVA, [Session 1]: $F(3, 57) = 56.40, P < 0.0001$; [Session 2]: $F(3, 57) = 31.82, P < 0.0001$; [Session 3]: $F(3, 57) = 30.49, P < 0.0001$; [Session 4]: $F(3, 57) = 33.63, P < 0.0001$, Tukey's multiple comparisons test, $**P < 0.01$ vs Pre CS (ITI). **(c)** The CS tones induced an increased poking time in both training box and novel box. $n = 20$ mice per group. Paired t test, [Training box]: $t = 18.7, df = 19, P < 0.0001$; [Novel box]: $t = 15.38, df = 19, P < 0.0001$. $**P < 0.01$ vs No CS. **(d)** Schematic for ambiguous cue test. **(e, f)** The CS-trained mice showed fewer poking times and longer first poking latency in response to a novel tone (5000 Hz, ambiguity) than to the CS tone (2000 Hz). $n = 20$ mice per group. Paired t test, [Poking times]: $t = 4.876, df = 19, P < 0.001$; [First poking latency]: $t = 3.370, df = 19, P = 0.0032$. $**P < 0.01$ vs CS tone. **(g)** The mice were further divided into more positive interpretation (MPI) and less positive interpretation (LPI) groups according to their different poking times (cutoff: 5 poking times) to the ambiguous tone (**left**), and both MPI and LPI groups had same poking times to the CS tones (**right**). $n = 11$ or 9 mice per group. Unpaired t test, [Novel tones]: $t = 6.959, df = 18, P < 0.0001$; [CS tones]: $t = 0.3927, df = 18, P = 0.6996$. $**P < 0.01$ vs MPI. **(h, i)** The MPI and LPI mice showed distinct activation patterns in the pBLA and the other indicated brain regions measured by c-Fos staining. Scale bar, $100 \mu\text{m}$. $n = 4\sim 5$ mice per group. Unpaired t test, [IL]: $t = 3.056, df = 8, P = 0.0157$; [pBLA]: $t = 2.683, df = 8, P = 0.0278$; [BMP]: $t = 2.672, df = 6, P = 0.0369$. $*P < 0.05$ vs LPI. **(j-l)** Activation of pBLA^{Calb1+} neurons was more significant in MPI than LPI group measured by c-Fos (red) and Calb1 (green) co-staining. Scale bar, $100 \mu\text{m}$. $n = 5$ to 7 mice per group. Unpaired t test, $t = 2.440, df = 10, P = 0.0348$ (k); Unpaired t test, $t = 2.999, df = 10, P = 0.0134$ (l). $*P < 0.05$ vs LPI. Data were presented as mean \pm SEM.

-The simplicity of the visual schematics (e.g. fig 5 a, e, i) relative to the complexity of the combined optogenetic/chemogenetic approaches does not capture nuance of the approach.

[Response]: Thanks to the reviewer. As suggested, we modified the visual schematics of Figure 5 in the revised version.

Figure 5. Activating IL-pBLA^{Calb1} circuit ameliorates anxiety and depression behaviors in CUMS mice. (a) Schematic of IL-pBLA photostimulation and pBLA^{Calb1} inhibition in CUMS mice during OFT, EPM and SPT. AAV-CaMKII-ChR2 and AAV-DIO-hM4Di were injected into the IL and pBLA of Calb1-Cre mice, respectively. At the 5th week of CUMS training, fibers were implanted in the pBLA. 30 min before behavior tests, CNO was intraperitoneally injected. ChR2-CUMS mice were tested in a single 9 min session in OFT or EPM with three 3-min epochs. (b, c,

e-g, h) Chr2-CUMS mice showed less anxiety and depression behaviors during the illumination epoch, evidenced by greater center entries (**b**) and time staying in the center (**c**) in OFT, more open-arm exploration (**e, f**) and shorter open-arm entry latency (**g**) in EPM, and higher sucrose preference in SPT (**h**), while these anti-anxiety and anti-depression effects were reversed when Calb1+ neurons in the pBLA were inhibited by CNO injection. No difference on travelled distance was detected during and after light stimulation or CNO injection (**d**). Two-way ANOVA group \times epoch interaction, n=11 mice per group. In OFT: [Center entries]: $F(4,90) = 3.971, P = 0.0052$; [Time in center]: $F(4,90) = 5.510, P = 0.0005$; [Distance moved]: $F(2,60) = 0.3377, P = 0.7174$. In EPM: [Open arm entries]: $F(4,75) = 2.581, P = 0.0439$; [Time in open arm]: $F(4,81) = 9.084, P < 0.0001$; [open-arm entry latency]: $F(4,75) = 3.344, P = 0.0142$. [In SPT]: $F(2,48) = 34.86, P < 0.0001$. Bonferroni post hoc analysis, $***P < 0.01$ vs eYFP:IL-pBLA+veh:pBLA; $### P < 0.01$ vs Chr2:IL-pBLA+veh:pBLA. Data were presented as mean \pm SEM. **(i)** Proposed working model. The excitatory IL-pBLA^{Calb1} circuit drives interpretation for ambiguous information in a bidirectional and reversible manner. Inhibiting IL-pBLA^{Calb1} pathway produces pessimistic phenotypes, while activating the circuit promotes positive interpretations for the ambiguity, exerting anti-anxiety and anti-depression effects.

- Please include a schematic of the CUMS paradigm/ control in the main figures

[Response]: As suggested, we have drawn a schematic to clarify the CUMS paradigm (please see Figure 4 and sTable 1).

Figure 4. CUMS induces anxiety and depression behaviors with IL-pBLA pathway inhibition.

(a) Scheduling for Chronic Unpredictable Mild Stress (CUMS). (b, e, h) Schematic of open field test (OFT, b), elevated plus maze (EPM, e) and sucrose preference test (SPT, h). (c, d) Compared with controls, CUMS mice showed fewer center entries and less time staying in the center during OFT. $n = 14$ mice per group. Unpaired t test, [Center entries] $t = 4.926$, $df = 26$, $P < 0.0001$; [Time in center] $t = 2.997$, $df = 26$, $P = 0.0059$. (f, g) In EPM, the CUMS mice showed a decreased open arm entry (f) and an increased entry latency to the open arm (g). $n = 14$ mice per group. Unpaired t test, [Open arm entries] $t = 3.868$, $df = 26$, $P = 0.0007$; [Open-arm entry latency] $t = 2.996$, $df = 26$, $P = 0.0059$. (i) In SPT, the CUMS mice showed less sucrose preference than the control mice. $n = 10$ mice per group. Unpaired t test, $t = 10.30$, $df = 18$, $P < 0.0001$.

n=10 mice per group. **(j)** Representative co-staining of CTB+/c-Fos+ neurons in the IL shown by pBLA injection of CTB (red) and measured at 90 min after EPM. Scale bar=100 μ m. **(k)** The percentage of both CTB+ and c-Fos+ over CTB+ was significantly decreased in CUMS group than controls. n = 6 mice per group. Unpaired *t* test, *t* = 9.341, *df* =10, *P*<0.0001. ***P*<0.01 vs Control. Data were presented as mean \pm SEM.

sTable 1

Week	Day	Food deprivation	Water deprivation	Foreign object exposure	Cold exposure	Illumination	Restraint
1	1	24h (6:00 a.m.-6:00 a.m. on day 2)	24h (6:00 a.m.-6:00 a.m. on day 2)				
	2			1 h (5:00 a.m.-6:00 a.m.)			2 h (6:00 p.m.-8:00 p.m.)
	3				2 h (9:00 a.m.-11:00 a.m.)	12 h (6:00 p.m.-6:00 a.m. on day 4)	
	4				2 h (7:00 a.m.-9:00 a.m.)		2 h (4:00 p.m.-6:00 p.m.)
	5	24 h (6:00 a.m.-6:00 a.m. on day 6)		1 h (7:00 p.m.-8:00 p.m.)			
	6		24h (6:00 a.m.-6:00 a.m. on day 7)			12 h (7:00 a.m.-7:00 p.m.)	
	7			1 h (9:00 a.m.-10:00 a.m.)	2 h (4:00 p.m.-6:00 p.m.)		

Week	Day	Food deprivation	Water deprivation	Foreign object exposure	Cold exposure	Illumination	Restraint
2	1	24h (6:00 a.m.-6:00 a.m. on day 2)					2 h (6:00 p.m.-8:00 p.m.)
	2		24h (6:00 a.m.-6:00 a.m. on day 3)			12 h (6:00 p.m.-6:00 a.m. on day 3)	
	3			1 h (5:00 a.m.-6:00 a.m.)	2 h (9:00 a.m.-11:00 a.m.)		2 h (4:00 p.m.-6:00 p.m.)
	4	24 h (6:00 a.m.-6:00 a.m. on day 5)		1 h (7:00 p.m.-8:00 p.m.)			
	5				2 h (7:00 a.m.-9:00 a.m.)		2 h (4:00 p.m.-6:00 p.m.)
	6		24h (6:00 a.m.-6:00 a.m. on day 7)	1 h (4:00 p.m.-5:00 p.m.)			2 h (8:00 p.m.-10:00 p.m.)
	7				2 h (4:00 p.m.-6:00 p.m.)	12 h (6:00 a.m.-6:00 p.m.)	2 h (8:00 a.m.-10:00 a.m.)

Week	Day	Food deprivation	Water deprivation	Foreign object exposure	Cold exposure	Illumination	Restraint
3	1		24h (6:00 a.m.-6:00 a.m. on day 2)				2 h (6:00 p.m.-8:00 p.m.)
	2	24h (6:00 a.m.-6:00 a.m. on day 3)		1 h (5:00 a.m.-6:00 a.m.)	2 h (4:00 p.m.-6:00 p.m.)		
	3					12 h (6:00 p.m.-6:00 a.m. on day 4)	2 h (4:00 p.m.-6:00 p.m.)
	4		24h (6:00 a.m.-6:00 a.m. on day 5)				2 h (4:00 p.m.-6:00 p.m.)
	5			1 h (7:00 a.m.-8:00 a.m.)	2 h (4:00 p.m.-6:00 p.m.)	12 h (6:00 a.m.-6:00 p.m.)	
	6	24h (6:00 a.m.-6:00 a.m. on day 7)			2 h (9:00 a.m.-11:00 a.m.)		2 h (3:00 p.m.-5:00 p.m.)
	7			1 h (1:00 p.m.-2:00 p.m.)	2 h (4:00 p.m.-6:00 p.m.)		2 h (8:00 a.m.-10:00 a.m.)

Supplemental Table 1: Experimental Stressors and Scheduling for Chronic

Unpredictable Mild Stress (CUMS). Mice were exposed to a variety of mild unexpected stressors for 6 weeks, including water and/or food deprivation, foreign object exposure, restraint, cold exposure, and reversal of the light/dark cycle. The procedures were repeated from week 1 after first 3 weeks. Some stressors were interrupted or modified when they occurred at particular points in the sucrose-preference test schedule.

- Please include an overview of placements across subjects for all experiments

[Response]: As suggested, we have made Supplemental Table 2 to overview the placements across subjects for all experiments (please see Key Resources sTable2).

REAGENT or RESOURCE	SOURCE	IDENTIFIER
Bacterial and Virus Strains		
HSV-EGFP	BrainVTA	Cat# H01001
rAAV2/R-hSyn-Cre-WPRE-hGH-PA	BrainVTA	Cat# PT-0136
rAAV2/9-CaMKIIa-DIO-GCaMP6f-EGFP	BrainVTA	Cat# PT-110-3-1
rAAV2/9-CaMKIIa-DIO-EGFP	BrainVTA	Cat# PT-0119
pAAV2/8-CaMKIIa-hChr2 (H134R)-eYFP	OBio	N/A
pAAV2/8-CaMKIIa-eNpHR3.0-eYFP	OBio	N/A
pAAV2/8-CaMKIIa-eYFP	OBio	N/A
pAAV2/8-CaMKIIa-hM3D (Gq)-eYFP	OBio	N/A
pAAV2/8-CaMKIIa-hM4D (Gi)-eYFP	OBio	N/A
pAAV2/8-CaMKIIa-DIO-hM3D (Gq)-eYFP	OBio	N/A
pAAV2/8-CaMKIIa-DIO-hM4D (Gi)-eYFP	OBio	N/A
pAAV2/8-CaMKIIa-DIO-eYFP	OBio	N/A
pAAV2/8-CaMKIIa-DIO-hChr2 (H134R)-eYFP	OBio	N/A
pAAV2/8-CaMKIIa-DIO-eNpHR3.0-eYFP	OBio	N/A
Antibodies		
Anti-c-Fos antibody	Synaptic Systems	Cat# 226 004; RRID:AB_2619946
Calbindin D-28k	Swant	Cat# 300; RRID:AB_10000347
Calbindin antibody [EP3478]	Abcam	Cat# ab108404; RRID:AB_10861236
Experimental Models:		
Organisms/Strains		
C57BL/6NJ mice	Center of Beijing Laboratory Technology	
Calb1-IRES2-Cre-D mice	Xiaohui Zhang lab	
Ai9 mice	Xiaohui Zhang lab	
Chemicals		
Clozapine N-oxide (CNO)	Sigma	Cat# C0832
DAPI	Beyotime	Cat# C1002
CTB555	BrainVTA	Cat# CTB-02

Reference

1. Ambroggi, F., Ishikawa, A., Fields, H.L. & Nicola, S.M. Basolateral amygdala neurons facilitate reward-seeking behavior by exciting nucleus accumbens neurons. *Neuron* **59**, 648–661 (2008).
2. Steinberg, E.E., *et al.* Amygdala-Midbrain Connections Modulate Appetitive and Aversive Learning. *Neuron* **106**, 1026–1043 e1029 (2020).
3. Goshen, I., *et al.* Brain interleukin-1 mediates chronic stress-induced depression in mice via adrenocortical activation and hippocampal neurogenesis suppression. *Molecular psychiatry* **13**, 717–728 (2008).
4. Zhou, Q.G., *et al.* Hippocampal telomerase is involved in the modulation of depressive behaviors. *The Journal of neuroscience : the official journal of the Society for Neuroscience* **31**, 12258–12269 (2011).
5. Sobrian, S.K., Marr, L. & Ressler, K. Prenatal cocaine and/or nicotine exposure produces depression and anxiety in aging rats. *Progress in neuro-psychopharmacology & biological psychiatry* **27**, 501–518 (2003).
6. Huang, H.J., *et al.* Ghrelin alleviates anxiety- and depression-like behaviors induced by chronic unpredictable mild stress in rodents. *Behavioural brain research* **326**, 33–43 (2017).
7. Liu, M.Y., *et al.* Sucrose preference test for measurement of stress-induced anhedonia in mice. *Nature protocols* **13**, 1686–1698 (2018).
8. Stuber, G.D., *et al.* Excitatory transmission from the amygdala to nucleus accumbens facilitates reward seeking. *Nature* **475**, 377–380 (2011).
9. Burgos-Robles, A., *et al.* Amygdala inputs to prefrontal cortex guide behavior amid conflicting cues of reward and punishment. *Nature neuroscience* **20**, 824–835 (2017).
10. Hu, R.K., *et al.* An amygdala-to-hypothalamus circuit for social reward. *Nature neuroscience* **24**, 831–842 (2021).
11. Namburi, P., *et al.* A circuit mechanism for differentiating positive and negative associations. *Nature* **520**, 675–678 (2015).
12. Vorhees, C.V. & Williams, M.T. Morris water maze: procedures for assessing spatial and related forms of learning and memory. *Nature protocols* **1**, 848–858 (2006).
13. Deacon, R.M. & Rawlins, J.N. T-maze alternation in the rodent. *Nature protocols* **1**, 7–12 (2006).
14. Oomen, C.A., *et al.* The touchscreen operant platform for testing working memory and pattern separation in rats and mice. *Nature protocols* **8**, 2006–2021 (2013).
15. Li, X., *et al.* Serotonin receptor 2c-expressing cells in the ventral CA1 control attention via innervation of the Edinger-Westphal nucleus. *Nature neuroscience* **21**, 1239–1250 (2018).
16. Walf, A.A. & Frye, C.A. The use of the elevated plus maze as an assay of anxiety-related behavior in rodents. *Nature protocols* **2**, 322–328 (2007).
17. Kalueff, A.V., *et al.* The regular and light-dark Suok tests of anxiety and sensorimotor integration: utility for behavioral characterization in laboratory rodents. *Nature protocols* **3**, 129–136 (2008).
18. Porsolt, R.D., Bertin, A. & Jalfre, M. "Behavioural despair" in rats and mice: strain differences and the effects of imipramine. *European journal of pharmacology* **51**, 291–294

(1978).

19. Todd, W.D., *et al.* A hypothalamic circuit for the circadian control of aggression. *Nature neuroscience* **21**, 717–724 (2018).
20. Harding, E.J., Paul, E.S. & Mendl, M. Animal behaviour: cognitive bias and affective state. *Nature* **427**, 312 (2004).
21. Waszczuk, M.A., Coulson, A.E., Gregory, A.M. & Eley, T.C. A longitudinal twin and sibling study of the hopelessness theory of depression in adolescence and young adulthood. *Psychological medicine* **46**, 1935–1949 (2016).
22. Plomin, R., *et al.* Optimism, pessimism and mental health: a twin/adoption analysis. *Pers Individ Dif* **13**, 921–930 (1992).
23. Carver, C.S., Scheier, M.F. & Segerstrom, S.C. Optimism. *Clinical psychology review* **30**, 879–889 (2010).
24. Resnik, J. & Paz, R. Fear generalization in the primate amygdala. *Nature neuroscience* **18**, 188–190 (2015).
25. Dymond, S., Dunsmoor, J.E., Vervliet, B., Roche, B. & Hermans, D. Fear Generalization in Humans: Systematic Review and Implications for Anxiety Disorder Research. *Behavior therapy* **46**, 561–582 (2015).
26. Bravo-Rivera, C., Roman-Ortiz, C., Brignoni-Perez, E., Sotres-Bayon, F. & Quirk, G.J. Neural structures mediating expression and extinction of platform-mediated avoidance. *The Journal of neuroscience : the official journal of the Society for Neuroscience* **34**, 9736–9742 (2014).
27. Diehl, M.M., *et al.* Divergent projections of the prelimbic cortex bidirectionally regulate active avoidance. *eLife* **9** (2020).
28. Xie, L., *et al.* Neonatal sevoflurane exposure induces impulsive behavioral deficit through disrupting excitatory neurons in the medial prefrontal cortex in mice. *Translational psychiatry* **10**, 202 (2020).

REVIEWER COMMENTS

Reviewer #1 (Remarks to the Author):

The manuscript is much improved and all comments have been addressed

Reviewer #2 (Remarks to the Author):

The authors have attempted to make improvements to the manuscript, but there are many issues still looming that make this paper, in my opinion, unsuitable for publication.

The major, big picture issue of the original version was over anthropomorphic interpretations of the findings. The data themselves (the statistics) are fine, but the interpretation is inappropriate. Rodents are not humans and thus should not be discussed as such. The authors have updated some of the language, but still use inappropriately anthropomorphized "optimist" and "pessimist" as well as "more/less positive interpretation". Respectfully, here are the reasons why you cannot do this:

1 "Positive interpretation" data in figure 1 are purely showing generalization. This is a well-documented, well-described phenomenon. The "more positive" group generalizes auditory cues more. There is no reason to rename generalization and ascribe human characteristics to it. To this point, these very same findings could actually reflect worse learning in the MPI group and thus be called "smart" versus "not so smart" groups. For the task and data in figure 1, it's only appropriate to use behavioral descriptors like 'generalization'.

2 The National Institutes of Mental Health (and our field in general) are moving away from the practice of over-humanizing animal behavior findings because it does not move our field forward, and can even harm future directions of research. This includes describing assays such as EPM, OFT, etc as models of depression or anxiety. Yes, one could cite 10+ years old papers (as done here) but that doesn't reflect the current state of our field. I am not saying these assays aren't valid in some regard, but they can't be described with human terminology.

NIMH statement here: <https://grants.nih.gov/grants/guide/notice-files/NOT-MH-19-053.html>

Based on the language used throughout the manuscript (all terms used, not just the ones above) and the odd and often very old citations, it seems that one major problem here could be a lack of thorough knowledge of the field and its literature. For example, there are statements about how no one has studied 'positive' aspects of ambiguous learning alongside negative. This is simply not true. (see the many papers by the Sangha group on discriminating safety/fear cues, for example). This is an issue that won't be easily remedied, but it needs to be done to be relevant to findings in the field.

Other troubling things not addressed:

- Sample images of histology are not enough; we need to see maps of viral spread for each group, especially because the authors are parsing anterior/posterior BLA and claiming IL injections never diffuse to PL. This is standard for our field.
- Line 517 states that on certain trials sucrose delivery was manually stopped by turning off the pump. How does one do this and ensure no mistakes are made? Does this mean someone stood by the operant box and watched each animal for each session?

- Fig 2C, the Y axis is an example of odd use of terms and is not clear what's being measured/depicted
- Fig 2g and 2i: colors for cfos and calb-1 are swapped. was this on purpose?
- for viral tracing, volumes of injections not listed
- often in the methods, important time-sensitive things are described in vague terms, like "CNO was injected ~30 min prior to test" and for tracing studies mice were euthanized "after 1 or 2 days" following injection. Timing/volume for virus affects expression.

Reviewer #3 (Remarks to the Author):

Thank you to the authors for addressing a number of my concerns. The re-written text and additional details are very helpful, and have dramatically improved the manuscript. I have some additional comments to help clarify the main findings.

1. The fact that neural activity in IL-to-pBLACalb neurons significantly increases in NP as well as PP mice is important. This finding is not brought up again in the discussion and the participation of this pathway in NP processing shouldn't be ignored (e.g. pg. 13, lines 296-7), even if it makes the story less simple. There is a different time course and strength of activation in NP mice, as seen in slope and peak (3I-p), which are likely meaningful, but this should be acknowledged, especially since the title and main message of the manuscript is about the fact that this pathway is selective for PP only, and it is likely more complex than that. Given that the authors don't inhibit this pathway in NP mice to check for effects on NP preference, it's hard to say whether the activity is functionally meaningful (perhaps less so than for the PP animals), or if it isn't involved at all, given that this pathway does activate for the NP animals as well as the PP animals.

2. In addition to the MPI/LPI interpretation of the data shown in figure 1, this data also can be viewed as showing a generalization response. First the animals are trained for over two weeks to associate a 2kHz tone with reward, and then they hear a 5kHz tone, and some animals show a faster generalized poking response to this new tone whereas others show a slower generalized response. It would be helpful to acknowledge that this is the case particularly given that now the authors have included generalization in the discussion.

3. In the Results section (pg. 6, line 116), the authors refer to the "no CS group". As the authors clarified in the rebuttal to my original comment about this, there is no separate "no CS group", but rather there is one group of animals that are being compared when no CS is played and when the CS is played. Please adjust the text to reflect this.

Please find below a point-by-point reply (in black) to the reviewers' comments (in blue):

Reviewer #1 (Remarks to the Author):

The manuscript is much improved and all comments have been addressed

[Response]: We sincerely appreciate the reviewer's positive comments on our manuscript.

Reviewer #2 (Remarks to the Author):

The authors have attempted to make improvements to the manuscript, but there are many issues still looming that make this paper, in my opinion, unsuitable for publication.

The major, big picture issue of the original version was over anthropomorphic interpretations of the findings. The data themselves (the statistics) are fine, but the interpretation is inappropriate. Rodents are not humans and thus should not be discussed as such. The authors have updated some of the language, but still use inappropriately anthropomorphized "optimist" and "pessimist" as well as "more/less positive interpretation". Respectfully, here are the reasons why you cannot do this:

1 "Positive interpretation" data in figure 1 are purely showing generalization. This is a well-documented, well-described phenomenon. The "more positive" group generalizes auditory cues more. There is no reason to rename generalization and ascribe human characteristics to it. To this point, these very same findings could actually reflect worse learning in the MPI group and thus be called "smart" versus "not so smart" groups. For the task and data in figure 1, it's only appropriate to use behavioral descriptors like 'generalization'.

2 The National Institutes of Mental Health (and our field in general) are moving away from the practice of over-humanizing animal behavior findings because it does not move our field forward, and can even harm future directions of research. This includes describing assays such as EPM, OFT, etc as models of depression or anxiety. Yes, one could cite 10+ years old papers (as done here) but that doesn't reflect the current state of our field. I am not saying these assays aren't valid in some regard, but they can't be described with human terminology.

NIMH statement here: <https://grants.nih.gov/grants/guide/notice-files/NOT-MH-19-053.html>

Based on the language used throughout the manuscript (all terms used, not just the ones above) and the odd and often very old citations, it seems that one major problem here could be a lack of thorough knowledge of the field and its literature. For example, there are statements about how no one has studied 'positive' aspects of ambiguous learning alongside negative. This is simply not true. (see the many papers by the Sangha group on discriminating safety/fear cues, for example). This is an issue that won't be easily remedied, but it needs to be done to be relevant to findings in the field.

[Response]: We deeply appreciate the reviewer's comments and suggestions. By reading the literature recommended by the reviewer, we have a better understanding of the field. We fully agreed with the reviewer that over anthropomorphic interpretations of the findings should be avoided in the scientific research. Therefore, we thoroughly revised the manuscript and corrected inappropriate descriptions, such as “positive emotional cognition”→“reward generalization”; “positive interpretation”→ “reward-generalized approaching”; “negative emotional cognition” → “aversive generalization”; “negative interpretation” → “aversion-generalized avoidance”. Furthermore, we supplemented the knowledge of generalization and refined the framing of the manuscript. We highlight that the main scientific issue of the present study is to explore whether and how the distinct pBLA neurons with their associated circuits necessarily lead to reward generalization and promote stress resilience to rescue anxiety and depression. Please see the red part in the revised version. We hope the reviewer will agree that the manuscript have been dramatically improved and is now acceptable for publication.

Other troubling things not addressed:

- Sample images of histology are not enough; we need to see maps of viral spread for each group, especially because the authors are parsing anterior/posterior BLA and claiming IL injections never diffuse to PL. This is standard for our field.

[Response]: We thank the reviewer for pointing out this issue. As suggested, we have supplemented images of virus injection to clarify the targeting precision. As images

showed, most viruses have been confined to the posterior BLA and IL (please see supplementary Figure 6, supplementary Figure 8).

Figure 6. Representative images of Chr2 expression in the pBLA. Most viruses have been confined to the pBLA. Scale bar, 500 μm .

Figure 8. Representative images of Chr2 expression in the IL. The expression of Chr2 was confined to the IL. Scale bar, 500 μm .

We paid special attention to the PL region as mentioned by the reviewer, which lies dorsal to IL and has been reported in fear expression and extinction (DeNardo, *et al. Nature neuroscience*, 2019). Our retrograde tracing results revealed that PL anatomically connected with pBLA (please Figure 3b). To explore the role of PL–pBLA circuit in reward generalization, we injected AAV-CaMKII-NpHR into the PL and stimulated its terminals in the pBLA of mice with high reward generalization (HRG) (please see supplementary Figure 10 and 11). The result showed no significant alterations in reward preference to the ambiguous cue, indicating a minor effect of PL–pBLA circuit on reward generalization. Combined with our findings in IL–pBLA circuit, we believe that IL–pBLA circuit, rather than PL–pBLA circuit, controls reward generalization in response to ambiguity.

Figure 10. Representative images of ChR2 expression in the PL. The expression of ChR2 was confined to the PL. Scale bar, 500 μ m.

Figure 11. PL-pBLA inputs has no effects on reward preference for the ambiguity. (a, d) Schematic showing light illuminating epoch when the intermediate tone was delivered during go/go paradigm. (b, e) Brief blue light pulses at 20 Hz precisely activated ChR2 cells (b), while yellow light illumination blocked evoked spiking of NpHR cells (e), in the pBLA. (c, f) Activating and inhibiting PL-pBLA inputs did not significantly change the first poking latency of ChR2-AP and NpHR-RP mice. Two-way ANOVA group \times epoch interaction; first poking latency [ChR2-AP] $F(1,36) = 0.9731$, $P = 0.3305$, Bonferroni post hoc analysis; [NpHR-RP] $F(1,36) = 0.01569$, $P = 0.9010$, Bonferroni post hoc analysis. $n = 10$ mice per group. Data were presented as mean \pm SEM.

- Line 517 states that on certain trials sucrose delivery was manually stopped by turning off the pump. How does one do this and ensure no mistakes are made? Does this mean someone stood by the operant box and watched each animal for each session?

[Response]: Thanks to the reviewer for this question. In the present study, staying on the platform with at least three limbs is recognized as a correct posture to the negative CS. It can be identified easily by the experimenter. If it presents upon positive tones, the experimenter will switch off the sucrose delivery system immediately. The complicated situation is that the animal briefly touched the platform (less than three limbs) and quickly turned back to poke for sucrose (turn-back latency is always less than 5 s). If it happened, it is quite hard for the experimenter to switch off the sucrose delivery system to avoid the remedial poking behaviour. Therefore, we just labeled it as a false response. At the end of FC and RC tests, the discrimination ability was evaluated. Training was continued until animals reached a stable baseline of correct discrimination responses on at least 70% of the trials (please see page 21 line 27 to 29).

- Fig 2C, the Y axis is an example of odd use of terms and is not clear what's being measured/depicted

[Response]: We sincerely appreciate the reviewer for pointing out this inappropriate expression. In the revised version, we replaced “Times of step-on” with “Times of jumps onto the platform”. Please see Figure 2c, d and sFigure 3c, e.

- Fig 2g and 2i: colors for cfos and calb-1 are swapped. was this on purpose?

[Response]: It was not on purpose. We also performed immunostaining with different colors for c-Fos and Calb1 (please see the data below). As it showed, number of c-Fos (green) and Calb1 (red) co-stained neurons was increased in the pBLA of RP group compared with AP group (a-c) with similar total Calb1-neuron numbers (d) which was consistent with the findings in Figure 2i-l.

Preliminary data. Number of c-Fos (green) and Calb1 (red) co-stained neurons was increased in the pBLA of RP group compared with AP group. No difference in the number of calb1 neurons was detected between AP and RP groups. Scale bar, 100 μ m. $n = 6$ mice per group. Unpaired t test, $t = 2.658$, $df = 10$, $P = 0.0240$ (b); $t = 3.678$, $df = 10$, $P = 0.0043$ (c); $t = 0.4028$, $df = 10$, $P = 0.5441$ (d). * $P < 0.01$ vs AP.

- for viral tracing, volumes of injections not listed

[Response]: We thank the reviewer very much. The volume of HSV injection for tracing is 100 nl. As suggested, we supplemented the details in the Method. Please see page 17 line 21.

- often in the methods, important time-sensitive things are described in vague terms, like "CNO was injected ~30 min prior to test" and for tracing studies mice were euthanized "after 1 or 2 days" following injection. Timing/volume for virus affects expression.

[Response]: We thank the reviewer very much. It was 30-32 min prior to test that CNO was injected. As suggested, we supplemented the details in the Method. Please see page 19 line 11 and page 17 line 23.

Reviewer #3 (Remarks to the Author):

Thank you to the authors for addressing a number of my concerns. The re-written text and additional details are very helpful, and have dramatically improved the manuscript. I have some additional comments to help clarify the main findings.

[Response]: We sincerely appreciate the reviewer's positive comments on our manuscript.

1. The fact that neural activity in IL-to-pBLA^{Calb1} neurons significantly increases in NP as well as PP mice is important. This finding is not brought up again in the discussion and the participation of this pathway in NP processing shouldn't be ignored (e.g. pg. 13, lines 296-7), even if it makes the story less simple. There is a different time course and strength of activation in NP mice, as seen in slope and peak (3l-p), which are likely meaningful, but this should be acknowledged, especially since the title and main message of the manuscript is about the fact that this pathway is selective for PP only, and it is likely more complex than that. Given that the authors don't inhibit this pathway in NP mice to check for effects on NP preference, it's hard to say whether the activity is functionally meaningful (perhaps less so than for the PP animals), or if it isn't involved at all, given that this pathway does activate for the NP animals as well as the PP animals.

[Response]: Many thanks to the reviewer for the question. As suggested by the Reviewer 2, we replaced positive (PP) and negative (NP) preference with reward (RP) and aversion preference (AP) respectively in the revised version. The population Ca²⁺ recording in Figure 3 was conducted on RP and AP mice only during their poking behaviours, not stepping-on behaviours, in response to the ambiguous cue. Compared with AP group, the average peak calcium activity and the slope for calcium transit before poke increased more significantly in the RP group (Figure 3n, o), indicating that the activation of the IL-pBLA circuit controls reward preference for the ambiguity, and that the more rapid and robust the activation of the IL-pBLA circuit is, the stronger higher reward preference for ambiguity. We added details (highlighted in red) to the

figure 3 legend.

To further address the reviewer's concern, we expressed the AAV8-CaMKIIa-eNpHR3.0-eYFP in the IL of AP mice. Photoinhibition IL-pBLA inputs had no effects on the stepping-on response in the presence of the first intermediate tone (sFigure 9), indicating its minor effects on aversive generalization. Together, these data demonstrated that the IL-pBLA circuit controls reward generalization for the ambiguity and triggers the reward-related approaching behaviours (page 15 line 14 to 18).

sFigure 9. Photoinhibiting IL-pBLA inputs had no effects on aversive preference in AP mice. (a) Schematic showing light illuminating epoch when the intermediate tone was delivered during go/go paradigm. (b) Yellow light illumination blocked evoked spiking of NpHR cells. (c) Inhibiting IL-pBLA inputs had no effect on the latency to step onto the platform in NpHR-AP mice. Two-way ANOVA group \times epoch interaction; [Latency to step onto the platform] $F(1,36) = 0.2240$, $P = 0.6389$, Bonferroni post hoc analysis. $n = 10$ mice per group. Data were presented as mean \pm SEM.

2. In addition to the MPI/LPI interpretation of the data shown in figure 1, this data also can be viewed as showing a generalization response. First the animals are trained for over two weeks to associate a 2kHz tone with reward, and then they hear a 5kHz tone, and some animals show a faster generalized poking response to this new tone whereas others show a slower generalized response. It would be helpful to acknowledged that this is the case particularly given that now the authors have included generalization in the discussion.

[Response]: We thank the reviewer very much for these constructive suggestions. We have refined the framing and supplemented the knowledge of generalization in the revised version. Please see the red part in the manuscript.

3. In the Results section (pg. 6, line 116), the authors refer to the “no CS group”. As the authors clarified in the rebuttal to my original comment about this, there is no separate “no CS group”, but rather there is one group of animals that are being compared when no CS is played and when the CS is played. Please adjust the text to reflect this.

[Response]: We apologize for our mistakes. In the revised version, we replaced “CS group” and “No CS group” with “CS” and “Pre-CS” respectively. Please see page 6 line 10 and Figure 1c. Thanks again for the reviewer for pointing out this mistake.